# Amino acid transporter SLC38A5 regulates developmental and pathological retinal angiogenesis

Zhongxiao Wang[1], Felix Yemanyi[1], Alexandra K Blomfield[1], Kiran Bora[1], Shuo Huang[1], Chi-Hsiu Liu[1], William R Britton[1], Steve S Cho[1], Yohei Tomita[1], Zhongjie Fu[1], Jian-xing Ma[2], Wen-hong Li[3], Jing Chen[1]*

[1]Department of Ophthalmology, Boston Children's Hospital, Harvard Medical School, Boston, United States; [2]Department of Biochemistry, Wake Forest University School of Medicine, Winston-Salem, United States; [3]Departments of Cell Biology and of Biochemistry, University of Texas Southwestern Medical Center, Dallas, United States

**Abstract** Amino acid (AA) metabolism in vascular endothelium is important for sprouting angiogenesis. SLC38A5 (solute carrier family 38 member 5), an AA transporter, shuttles neutral AAs across cell membrane, including glutamine, which may serve as metabolic fuel for proliferating endothelial cells (ECs) to promote angiogenesis. Here, we found that *Slc38a5* is highly enriched in normal retinal vascular endothelium, and more specifically, in pathological sprouting neovessels. *Slc38a5* is suppressed in retinal blood vessels from *Lrp5*$^{-/-}$ and *Ndp*$^{y/-}$ mice, both genetic models of defective retinal vascular development with Wnt signaling mutations. Additionally, *Slc38a5* transcription is regulated by Wnt/β-catenin signaling. Genetic deficiency of *Slc38a5* in mice substantially delays retinal vascular development and suppresses pathological neovascularization in oxygen-induced retinopathy modeling ischemic proliferative retinopathies. Inhibition of *SLC38A5* in human retinal vascular ECs impairs EC proliferation and angiogenic function, suppresses glutamine uptake, and dampens vascular endothelial growth factor receptor 2. Together these findings suggest that SLC38A5 is a new metabolic regulator of retinal angiogenesis by controlling AA nutrient uptake and homeostasis in ECs.

*For correspondence:
jing.chen@childrens.harvard.edu

**Competing interest:** The authors declare that no competing interests exist.

## Editor's evaluation

Anti-VEGF treatment is currently used to treat patients with pathological retinal angiogenesis, but finding the underlying cause of increased VEGF has been a challenge for the field. Wang and colleagues determined the role played of amino acid transporter, SLC38A5, in retinal angiogenesis and they showed that SLC38A5 in the retina is under the control of Wnt/β-catenin signaling. The deficiency of SLC38A5 resulted in delayed retinal vascular growth and reduced neovascularization in an Oxygen-Induced Retinopathy model. Additionally, the authors addressed the mechanisms of Slc38a5 as a glutamine transporter regulating retinal vascular development through VEGF receptors.

## Introduction

Angiogenesis, the growth of new blood vessels from existing vessels, is important in both development and disease (*Folkman, 1995*). In the developing eye, formation of blood vessels allows delivery of nutrients and removal of metabolic waste from neuronal retinas (*Selvam et al., 2018*; *Stahl et al., 2010*). In vascular eye diseases, specifically proliferative retinopathies, such as retinopathy of prematurity and diabetic retinopathy, abnormal proliferation of pathological blood vessels may lead to retinal

detachment and vision loss (*Chen and Smith, 2007*; *Hartnett and Penn, 2012*; *Antonetti et al., 2012*). Angiogenesis is coordinated by many pro-angiogenic factors, such as vascular endothelial growth factor (VEGF) and angiogenic inhibitors. The balance of these factors maintains vascular endothelium in proper homeostasis. In addition to growth factors, metabolism in vascular endothelial cells (ECs) has been recognized as a driving force of angiogenesis (*Rohlenova et al., 2018*; *De Bock et al., 2013a*). Both glucose glycolysis (*De Bock et al., 2013b*; *Yu et al., 2017*; *Wilhelm et al., 2016*) and fatty acid oxidation (*Vander Heiden et al., 2009*; *Neely and Morgan, 1974*; *Saddik and Lopaschuk, 1994*; *Kuo et al., 2017*) play major roles in regulating angiogenesis, and more specifically in ocular angiogenesis (*Joyal et al., 2017*; *Fu et al., 2017*; *Smith and Connor, 2005*). Moreover, metabolism of amino acids (AAs), such as glutamine, is increasingly established as an essential energy source in EC sprouting and angiogenesis (*Rohlenova et al., 2018*; *Dang, 2012*).

Several AAs may serve as a metabolic fuel for proliferating ECs to promote angiogenesis, including glutamine, asparagine, serine, and glycine (*Huang et al., 2017*; *Kim et al., 2017a*; *Zhang et al., 2014*; *Amelio et al., 2014*). Glutamine, the most abundant non-essential AA in the body, is a key carbon and nitrogen source and can be metabolized to sustain EC growth (*Huang et al., 2017*; *DeBerardinis and Cheng, 2010*). It can act as an anaplerotic source of carbon to replenish the tricarboxylic acid (TCA) cycle and support protein and nucleotide synthesis in EC growth (*Huang et al., 2017*; *Kim et al., 2017a*). Glutamine deprivation severely impairs EC proliferation and vessel sprouting, and pharmacological blockade of glutaminase 1, an enzyme that converts glutamine to glutamate, inhibits pathological angiogenesis (*Huang et al., 2017*). In addition to anaplerosis, glutamine also regulates angiogenesis by mediating AA synthesis, functioning as a precursor to other AAs such as glutamic acid, aspartic acid, and asparagine, as well as mediating macromolecule synthesis and redox homeostasis (*Huang et al., 2017*; *Kim et al., 2017a*; *Parfenova et al., 2006*; *van den Heuvel et al., 2012*; *Son et al., 2013*; *Parri and Chiarugi, 2013*). When glutamine is low, asparagine may be used as an alternative AA source (*Pavlova et al., 2018*) and can partially rescue glutamine-restricted EC defects (*Huang et al., 2017*; *Kim et al., 2017a*). In brain vascular ECs with barrier polarity, glutamine transport relies on facilitative transport systems on the luminal membrane of EC and sodium-dependent transport systems on the EC abluminal membrane, to actively transport glutamine from extracellular environment into ECs (*Lee et al., 1998*).

Solute carrier family 38 member 5 (SLC38A5, also known as SNAT5: system N sodium-coupled AA transporter 5) transports neutral AAs across cell membrane, including glutamine, asparagine, histidine, serine, alanine, and glycine (*Nakanishi et al., 2001*). Previously, SLC38A5 was found to mediate transcellular transport of AAs in brain glial cells (*Cubelos et al., 2005*). SLC38A5 is also a recently identified marker for pancreatic progenitors (*Stanescu et al., 2017*), where it is important for L-glutamine-dependent nutrient sensing and pancreatic alpha cell proliferation and hyperplasia (*Dean et al., 2017*; *Kim et al., 2017b*). In the eye, SLC38A5 was previously found in Müller glial cells and retinal ganglion cells (*Umapathy et al., 2005*; *Umapathy et al., 2008*), yet its localization and function in other retinal cells, including vascular ECs, are less clear. We and others previously found that *Slc38a5* was drastically down-regulated in both *Lrp5$^{-/-}$* and *Ndp$^{y/-}$* retinas (*Chen et al., 2012*; *Xia et al., 2010*; *Schäfer et al., 2009*), experimental models of two genetic vascular eye diseases: familial exudative vitreoretinopathy (FEVR) and Norrie disease, respectively. Both disease models have genetic mutations in Wnt signaling and share similar retinal vascular defects including initial incomplete or delayed vascular development, absence of deep retinal vascular layer, followed by a secondary hypoxia-driven increase in VEGF in the retina, and tuft-like neovascularization in the superficial layer of the retinal vasculature (*Ye et al., 2010*; *Benson, 1995*; *Wang et al., 2016*; *Andersen and Warburg, 1961*). *Slc38a5* is down-regulated 7–10-fold in *Lrp5$^{-/-}$* retinas and *Ndp$^{y/-}$* retinas during development (*Chen et al., 2012*; *Schäfer et al., 2009*), indicating a potential strong connection of *Slc38a5* with retinal blood vessel formation.

This study explored the regulatory functions of SLC38A5 in retinal angiogenesis during development and in disease. We found that *Slc38a5* expression is enriched in retinal blood vessels and its transcription is regulated by Wnt signaling. Moreover, genetic deficiency of *Slc38a5* impairs both developmental retinal angiogenesis and pathological retinal angiogenesis in a mouse model of oxygen-induced retinopathy (OIR), modeling proliferative retinopathies. Furthermore, inhibition of SLC38A5 decreases EC angiogenic function in vitro, and dampens EC glutamine uptake and growth factor signaling including VEGFR2. Together these findings identified a pro-angiogenic role of

SLC38A5 in ocular angiogenesis and suggest this transporter as a potential new target for developing therapeutics to treat pathological retinal angiogenesis.

## Results

### *Slc38a5* expression is enriched in retinal blood vessels and down-regulated in Wnt signaling deficient *Lrp5$^{-/-}$* and *Ndp$^{y/-}$* retinal vessels

We found that the mRNA levels of *Slc38a5* were consistently down-regulated in both *Lrp5$^{-/-}$* and *Ndp$^{y/-}$* retinas lacking Wnt signaling from postnatal day (P) 5 through P17, compared with their age-matched wild type (WT) controls (*Figure 1A and B*). To localize the cellular source of *Slc38a5,* blood vessels were isolated from retinal cross-sections by laser capture microdissection (LCM), followed by mRNA expression analysis using RT-qPCR. There was ~40-fold enrichment of *Slc38a5* mRNA level in retinal blood vessels compared with that in the whole retinas in WT mice (*Figure 1C*). Localization of SLC38A5 in retinal blood vessel endothelium is also demonstrated by immunohistochemistry showing colocalization of SLC38A5 with isolectin B$_4$, a marker of vascular endothelium (*Figure 1D*). Moreover, *Slc38a5* mRNA levels were substantially down-regulated in both *Lrp5$^{-/-}$* and *Ndp$^{y/-}$* LCM-isolated retinal vessels compared with their respective WT control blood vessels (*Figure 1E*). Protein levels of *Slc38a5* were also significantly decreased in both P17 *Lrp5$^{-/-}$* and *Ndp$^{y/-}$* retinas versus their respective WT control retinas, with more substantial reduction in *Ndp$^{y/-}$* retinas by ~50% (*Figure 1F and G*).

*Slc38a5* expression in retinal cells was further analyzed in single-cell transcriptome datasets of P14 C57B6 mouse retinas (*Macosko et al., 2015*), where *Slc38a5* is found mainly expressed in vascular endothelium, similar as endothelium marker *Pecam1* (*Figure 1—figure supplement 1*). In a human retinal single-cell dataset - Cell atlas of the human fovea and peripheral retina (*Yan et al., 2020*), *SLC38A5* is also mainly expressed in vascular endothelium, just like *PECAM1* (*Figure 1—figure supplement 1*).

Together these data demonstrate significant down-regulation of *Slc38a5* mRNA and protein levels in *Lrp5$^{-/-}$* and *Ndp$^{y/-}$* retinas, which is consistent with previous gene array findings from our and others' studies (*Chen et al., 2012*; *Schäfer et al., 2009*). These findings also strongly support vascular endothelium specificity of *Slc35a5* and its potential role in angiogenesis.

### Wnt signaling regulates *Slc38a5* transcription in vascular endothelium

To assess whether modulation of Wnt signaling regulates *Slc38a5* expression in retinal vascular endothelium, human retinal microvascular ECs (HRMECs) were cultured and treated with recombinant Norrin, a Wnt ligand, in the presence or absence of a Wnt inhibitor XAV939. XAV939 is a small molecule inhibitor of the Wnt signaling pathway, and it works through binding to tankyrase and stabilizing the Axin proteins, thus increasing degradation of β-catenin and blocking Wnt signaling (*Afifi et al., 2014*). We found that *Slc38a5* mRNA and protein levels were substantially induced by Norrin by ~1.6-fold (*Figure 2A and B*), whereas subsequent treatment with XAV939 reversed the Norrin-induced SLC38A5 upregulation in HRMECs (*Figure 2A and B*). Moreover, Wnt3a-conditioned medium (Wnt3a-CM) induced an even more potent upregulation of SLC38A5 by ~4-fold in mRNA levels and ~10-fold in protein levels (*Figure 2A and C*), which were also reversed by XAV939 treatment (*Figure 2A and C*). Activation or inhibition of Wnt signaling by Wnt modulators (Norrin, Wnt3a-CM, and XAV939) was confirmed in Western blot by the levels of the active β-catenin (non-phosphorylated-β-catenin), the canonical Wnt effector (*Figure 2B&C*).

To determine whether Wnt signaling regulates *SLC38A5* expression at the transcription level, dual-luciferase reporter assays were constructed in HEK293T cells. Canonical Wnt signaling mediates transcription of its target genes through recognizing β-catenin-responsive TCF-binding motifs (A/TA/TCAAAG) on their regulatory regions (*Oosterwegel et al., 1991*). Three *SLC38A5* promoter regions containing putative TCF-binding motifs were identified, cloned, and ligated with a luciferase reporter. All three luciferase reporter-containing promotor regions: P1, P2, and P3, showed significant increase in luciferase activity when co-transfected with active β-catenin, with both P1 and P2 showing ~three-fold increase and P3 showing ~twofold increase (*Figure 2D*). Together, these results suggest that SLC38A5 transcription is a downstream gene regulated by Wnt signaling via β-catenin binding to potentially multiple TCF-binding sites on its promoter regions, either directly or through an indirect secondary mechanisms via another Wnt responsive gene.

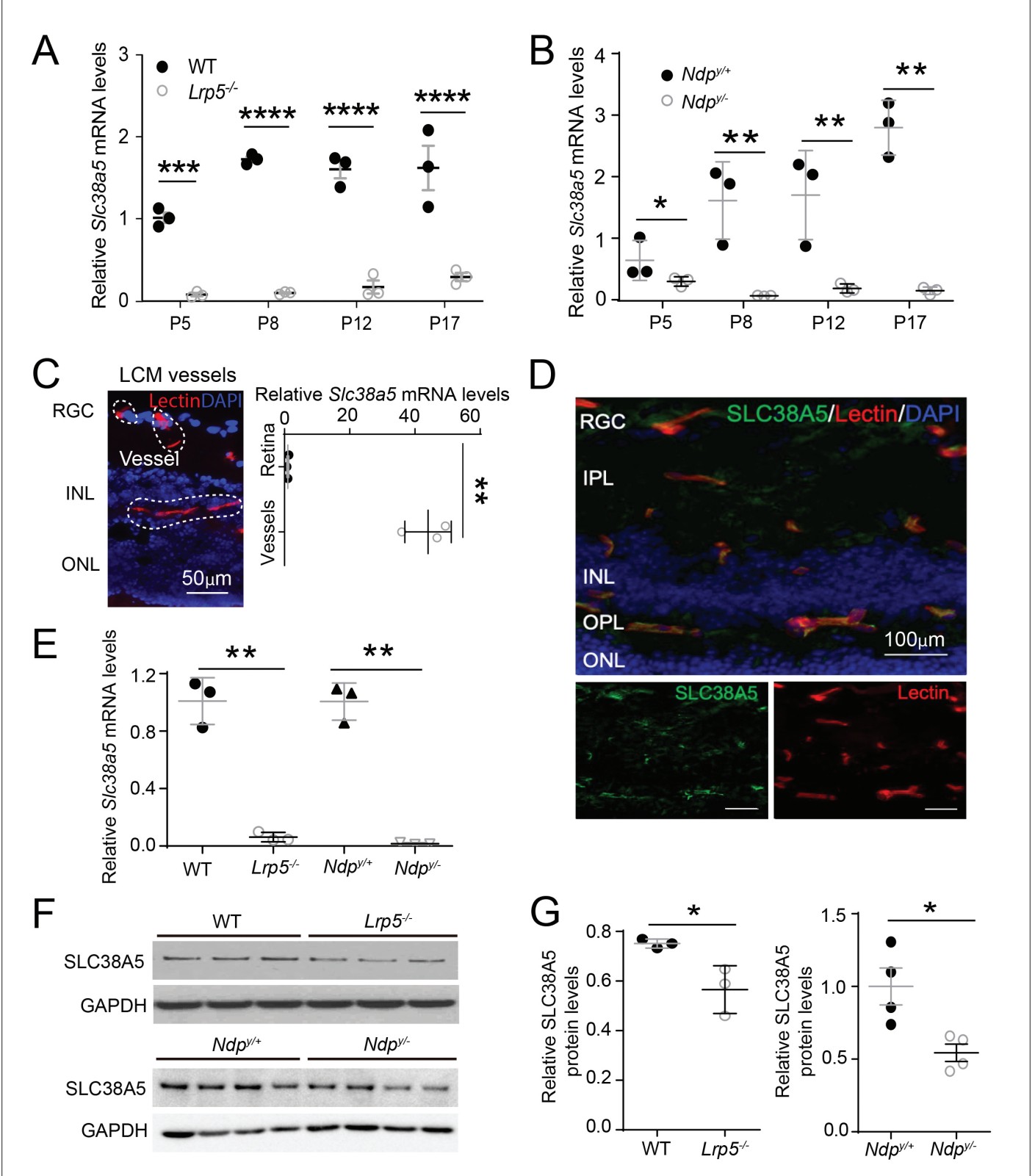

**Figure 1.** *Scl38a5* expression is enriched in retinal blood vessels and down-regulated in the retinas and retinal blood vessels of Wnt signaling deficient *Lrp5⁻/⁻* and *Ndpʸ/⁻* mice. (**A–B**) mRNA levels of *Slc38a5* were measured by RT-PCR in *Lrp5⁻/⁻* (**A**) and *Ndpʸ/⁻* (**B**) retinas compared with their respective wild type (WT) controls during development at postnatal day 5 (P5), P8, P12, and P17. (**C**) Retinal blood vessels stained with isolectin B₄ (red, outlined by the white dashed lines) were isolated by laser-captured microdissection (LCM) from retinal cross-sections. Cell nuclei were stained with DAPI

*Figure 1 continued on next page*

*Figure 1 continued*

(4',6-diamidino-2-phenylindole, blue) for illustration purpose only. LCM retinal samples were stained with only isolectin B$_4$ without DAPI. *Slc38a5* mRNA levels in LCM isolated retinal blood vessels were compared with the whole retinal levels using RT-qPCR. Scale bars: 50 µm. (**D**) Immunohistochemistry of retinal cross sections from WT eyes shows colocalization of SLC38A5 antibody (green) in retinal blood vessels stained with isolectin (red), counterstained with nuclear staining DAPI (blue). (**E**) *Slc38a5* mRNA levels in LCM-isolated *Lrp5*$^{-/-}$ and *Ndp*$^{y/-}$ retinal blood vessels were quantified with RT-qPCR and compared with their respective WT controls. (**F–G**) Protein levels of SLC38A5 (52 kDa) in P17 *Lrp5*$^{-/-}$ and *Ndp*$^{y/-}$ retinas and their WT controls were quantified with Western blot (**F**) and normalized by glyceraldehyde-3-phosphate dehydrogenase (GAPDH, 37 kDa) levels (**G**). RGC: retinal ganglion cells, IPL: inner plexiform layer, INL: inner nuclear layer, OPL: outer plexiform layer, ONL: outer nuclear layer. Data are expressed as mean ± SEM. n=3–4 per group. *p≤0.05, **p≤0.01, ****p≤0.0001.

The online version of this article includes the following source data and figure supplement(s) for figure 1:

**Source data 1.** Raw data for *Figure 1*.

**Figure supplement 1.** Distinct expression of *Slc38a5* in vascular endothelium in mouse and human retina with single-cell transcriptomics.

## Genetic deficiency of *Slc38a5* impairs developmental retinal angiogenesis

To evaluate the role of *Slc38a5* in retinal vessel development, we injected intravitreally siRNA targeting *Slc38a5* (si-*Slc38a5*) and control negative siRNA (si-Ctrl) to developing C57BL/6J mouse eyes and analyzed the impact on vascular development. Treatment with siRNA resulted in substantial reduction of *Slc38a5* mRNA levels by ~70% (n=3/group, p≤0.01, *Figure 3A*) and protein levels by ~70% (n=3/group, p≤0.05, *Figure 3B*) compared with contralateral si-Ctrl injected eyes at 3 days postinjection. Moreover, si-*Slc38a5* treated retinas showed significant (~20%) delay of superficial vascular coverage in the retinas at P7, 3 days after injection at P4, compared with si-Ctrl injection (n=9/group, p≤0.05, *Figure 3C*). In addition, the development of deep retinal vascular layer was also substantially impaired with almost 40% reduction in deep layer vascular area at P10, 3 days after siRNA were injected at P7 (n=11/group, p≤0.01, *Figure 3D*). These data suggest that knockdown of *Slc38a5* expression by siRNA delivery impedes retinal vascular development.

The role of *Slc38a5* in retinal vessel development was further evaluated in mutant mice with genetic deficiency of *Slc38a5*. Mice with targeted disruption of the *Slc38a5* gene are grossly normal yet have decreased pancreatic alpha cell hyperplasia induced by glucagon receptor inhibition (*Kim et al., 2017b*). Retinal blood vessel development was substantially delayed in *Slc38a5*$^{-/-}$ retinas for superficial vascular layer at P5 (n=8–11/group, p≤0.01, *Figure 3E*) and deep vascular layer at P10 (n=12–13/group, p≤0.01, *Figure 3F*). The delayed vascular development, however, is resolved in adult *Slc38a5*$^{-/-}$ mice when the retinal vasculature is largely normal in both superficial and deep layers (*Figure 3—figure supplement 1*) with comparable vessel density in vascular coverage and numbers of branches and junctions (*Figure 3—figure supplement 2*). These findings suggest that genetic knockout of *Slc38a5* results in delayed vascular development in the retinas and highlight an important role of *Slc38a5* in normal retinal vessel development.

## SLC38A5 is enriched in pathological neovessels and its genetic deficiency dampens pathological angiogenesis in OIR

To evaluate the role of *Slc38a5* in pathological retinal angiogenesis, we used a well-established OIR model (*Smith et al., 1994*), mimicking the hypoxia-induced proliferative phase as seen in retinopathy of prematurity and diabetic retinopathy. In this model, neonatal mice and their nursing mother were exposed to 75 ± 2% oxygen from P7 to P12 to induce vaso-obliteration, followed by return to room air, when relative hypoxia induces vaso-proliferation with maximal neovessel formation observable at P17. We found that *Slc38a5* mRNA expression was significantly suppressed during the vaso-obliteration phase between P8 and P12, and up-regulated during the vaso-proliferative phase of OIR between P14 and P17, compared with age-matched room air control mice (*Figure 4A*). Enrichment of *Slc38a5* in pathological neovessels was also quantitatively measured in OIR neovessels vs. normal vessels isolated using LCM. In pathological neovessels from P17 OIR retinas, *Slc38a5* mRNA levels were enriched at ~2.5-fold, compared with the levels in age-matched normoxic vessels (*Figure 4B*). Moreover, SLC38A5 protein levels were also increased at P17 OIR whole retinas by ~fourfold compared with age-matched normoxic retinas in Western blot (*Figure 4C*). These data indicate that *Slc38a5* is up-regulated in pathological retinal neovessels and suggestive of its angiogenic role in retinopathy.

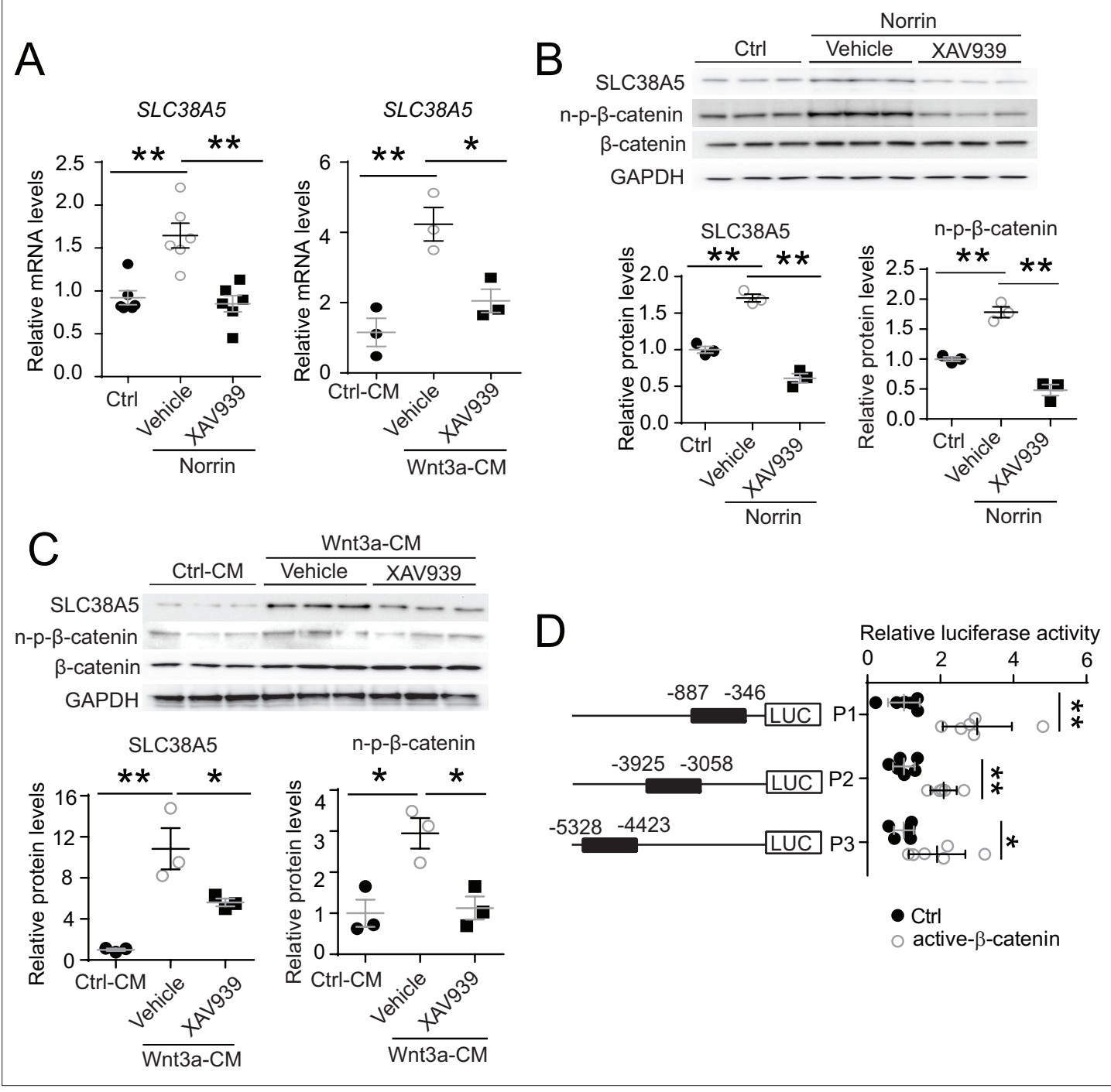

**Figure 2.** *Slc38a5* is a target gene of Wnt signaling in the vascular endothelium. (**A**) *Slc38a5* mRNA levels were increased in human retinal microvascular endothelial cells (HRMECs) treated with Wnt ligands, recombinant Norrin, and Wnt3a-conditioned medium (Wnt3a-CM), compared with their respective vehicle controls (Ctrl, and Ctrl-CM), and suppressed by a Wnt inhibitor XAV939. (**B–C**) Protein levels of SLC38A5 in HRMECs were up-regulated by Wnt ligands Norrin (**B**) and Wnt3a-CM (**C**), and down-regulated by XAV939. Protein levels of SLC38A5 (52 kDa) and β-catenin (92 kDa) were quantified by Western blotting and normalized by glyceraldehyde-3-phosphate dehydrogenase (GAPDH, 37 kDa) levels. n-p-β-catenin: non-phosphorylated β-catenin (92 kDa). (**D**) Three promoter regions upstream of *Slc38a5* gene containing potential Wnt-responsive TCF (T-cell factor)-binding motifs (TTCAAAG) was identified based on sequence analysis. Three putative TCF-binding regions: P1 (–887 bp to –346 bp), P2 (–3925 bp to –3058 bp), and P3 (–5328 bp to –4423 bp) were cloned and ligated separately with a luciferase reporter, and co-transfected with an active β-catenin plasmid in HEK 293T cells, followed by measurement of luciferase activity. Data are expressed as mean ± SEM. (**A-C**) n=3–6 per group. (**D**) n=5 per construct group. *p≤0.05, **p≤0.01.

The online version of this article includes the following source data for figure 2:

**Source data 1.** Raw data for *Figure 2*.

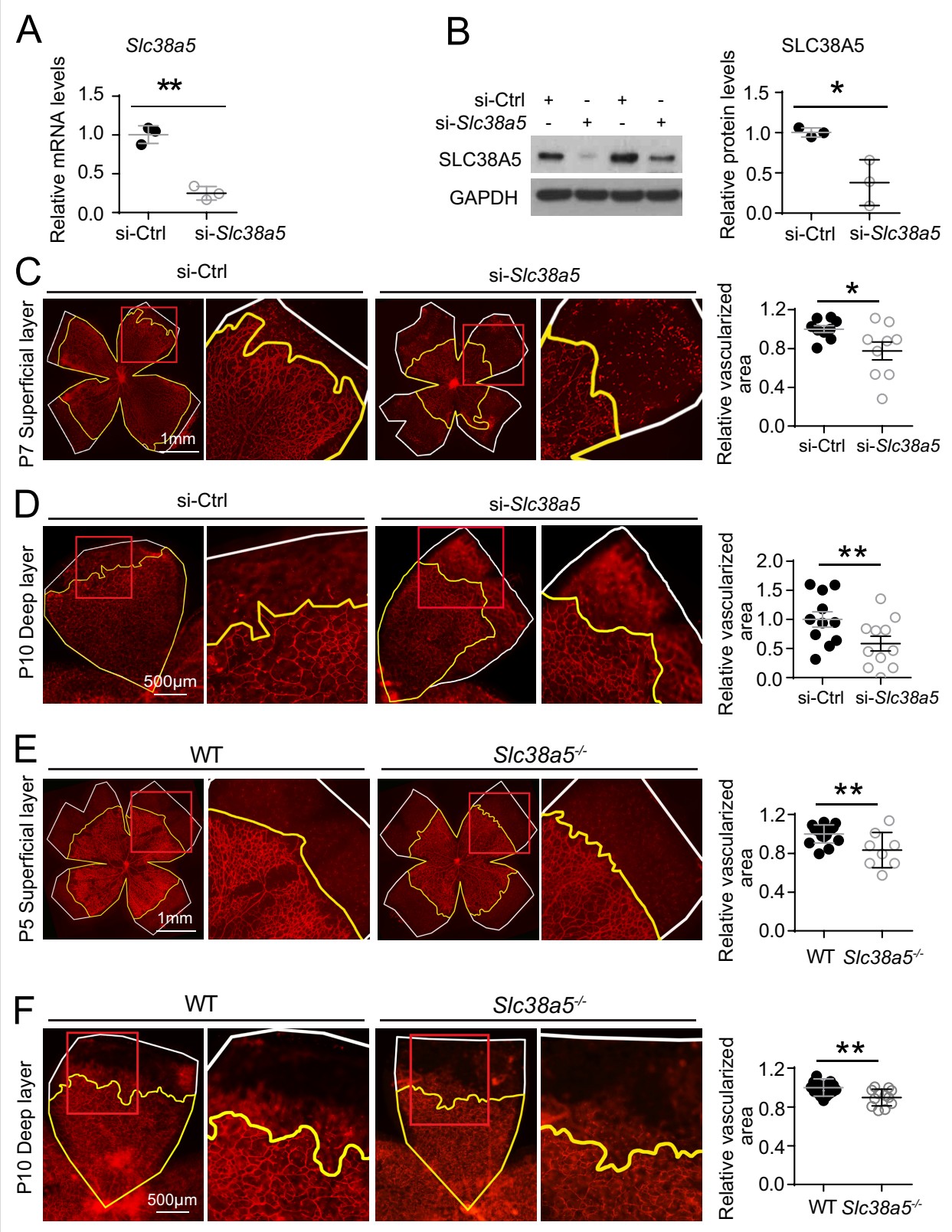

**Figure 3.** Genetic deficiency of *Slc38a5* impairs developmental retinal angiogenesis in vivo. (**A-D**) siRNA targeting *Slc38a5* (si-*Slc38a5*) was intravitreally injected in C57BL/6J mice , and the same volume of negative control siRNA (si-Ctrl) was injected into the contralateral eyes. Mice were sacrificed 3 days after injection, and retinas were isolated to detect expression level or to quantify vascular growth. mRNA (**A**) and protein (**B**) levels of SLC38A5 (52 kDa) confirm successful knockdown. Proteins were normalized to GAPDH (37 kDa). Each lane represents one retina. Retinal vascular coverage of superficial

*Figure 3 continued on next page*

*Figure 3 continued*

layer at P7 (**C**) and deep layer at P10 (**D**) was analyzed 3 days after intravitreal injection of si-*Slc38a5* and compared with their respective controls. Retinas were dissected, stained with isolectin B$_4$ (red), and then flat-mounted to visualize the vasculature. Percentages of vascularized area were quantified in superficial (C, n=9/group) or deep (D, n=11/group) vascular layer. (**E&F**) Retinal blood vessel development in *Slc38a5*$^{-/-}$ and WT littermate control mice from the same colony was imaged and quantified at P5 (E, n=8–11/group) and P10 (F, n=12–13/group), with staining of isolectin B$_4$ (red) to visualize the vasculature. In panels **C–F**, yellow lines outline retinal vascular areas, and white lines indicate total retinal areas. Red boxes indicate location of enlarged insets as shown on the right. Each dot represents one retina. Data are expressed as individual value and mean ± SEM. *p≤0.05, **p≤0.01.

The online version of this article includes the following source data and figure supplement(s) for figure 3:

**Source data 1.** Raw data for *Figure 3*.

**Figure supplement 1.** Adult *Slc38a5*$^{-/-}$ retinas appear normal with intact vascular barrier.

**Figure supplement 2.** Quantification of adult wild type (WT) and *Slc38a5*$^{-/-}$ retinas shows comparable retinal vascular density.

To further investigate the role of OIR-induced *Slc38a5* up-regulation in pathological neovessels, we subjected *Slc38a5*$^{-/-}$ mice to OIR. *Slc38a5*$^{-/-}$ mice showed significantly decreased levels of pathological neovascularization at P17 by ~25% compared with WT littermate controls (n=20/group, p≤0.01, *Figure 4D*), whereas the vaso-obliterated retinal areas were comparable at both P17 (*Figure 4D*) and P12 (n=10–12/group, *Figure 4—figure supplement 1*). These data suggest that loss of *Slc38a5* leads to significantly decreased levels of pathological neovascularization in OIR without impacting vessel loss.

## Modulation of SLC38A5 regulates EC angiogenic function in vitro

The function of SLC38A5 in mediating EC angiogenesis was assessed in HRMEC culture using siRNA to knockdown *SLC38A5*. Compared with negative control siRNA (si-Ctrl), *SLC38A5* siRNA (si-*SLC38A5*) effectively suppressed *SLC38A5* mRNA expression by more than 80% (p≤0.01, *Figure 5A*), and protein level by ~50% (p≤0.05, *Figure 5B*), confirming successful inhibition. EC viability and/or proliferation were analyzed with MTT (3-(4,5-dimethylthiazol-2-yl)-2,5-diphenyltetrazolium bromide) cellular metabolic activity assay (measuring NAD(P)H-dependent oxidoreductase activities) at 1–3 days after siRNA transfection. HRMEC metabolic activity was significantly decreased at 48 and 72 hr after transfected with si-*SLC38A5* (~30% reduction at 72 hr), compared with si-Ctrl-treated HRMECs (p≤0.01, *Figure 5C*), indicative of decreased cell viability and/or proliferation. In addition, si-*SLC38A5* substantially suppressed HRMEC migration (p≤0.05, *Figure 5D*) and tubular formation, resulting in ~35% reduction in total vessel length (*Figure 5E*). Together these results suggest that knockdown of AA transporter SLC38A5 suppresses EC angiogenic function.

## SLC38A5 inhibition decreases EC glutamine uptake

As an AA transporter, one of the main AAs that SLC38A5 transports is glutamine. Here, the impact of SLC38A5 on EC uptake of glutamine was measured with a glutamine/glutamate uptake bioluminescent assay in HRMECs. Suppression of SLC38A5 with si-*SLC38A5* resulted in ~25% decrease in the amount of intracellular glutamine measured by bioluminescence and then titrated to a standard concentration curve, compared with si-Ctrl treatment (p≤0.01, *Figure 6A*). Conversely, measurement of corresponding cell culture medium collected from si-*SLC38A5*-treated HRMECs also shows a reciprocal ~25% higher levels of glutamine content than si-Ctrl-treated cells (p≤0.01, *Figure 6B*), suggesting effective blockade of glutamine transport into HRMECs.

The effect of glutamine on HRMEC angiogenesis was then further examined with angiogenic assays. Treatment with glutamine enhanced HRMEC proliferation in MTT assays, which was blocked by a glutamine antagonist 6-diazo-5-oxo-L-norleucine (DON) (*Figure 6D*). DON has structural similarity with glutamine, and hence, can lead to competitive binding of glutamine-binding proteins and their irreversible inhibition by forming a covalent adduct (*Lemberg et al., 2018*). Moreover, glutamine treatment substantially increased HRMEC migration by ~50% (*Figure 6B and E*) and tubular formation by more than twofold (*Figure 6C and F*), both of which were reversed by DON treatment. These findings further suggest that loss of SLC38A5 may limit glutamine uptake and reduce its bioavailability in vascular EC, thereby leading to decreased EC angiogenesis.

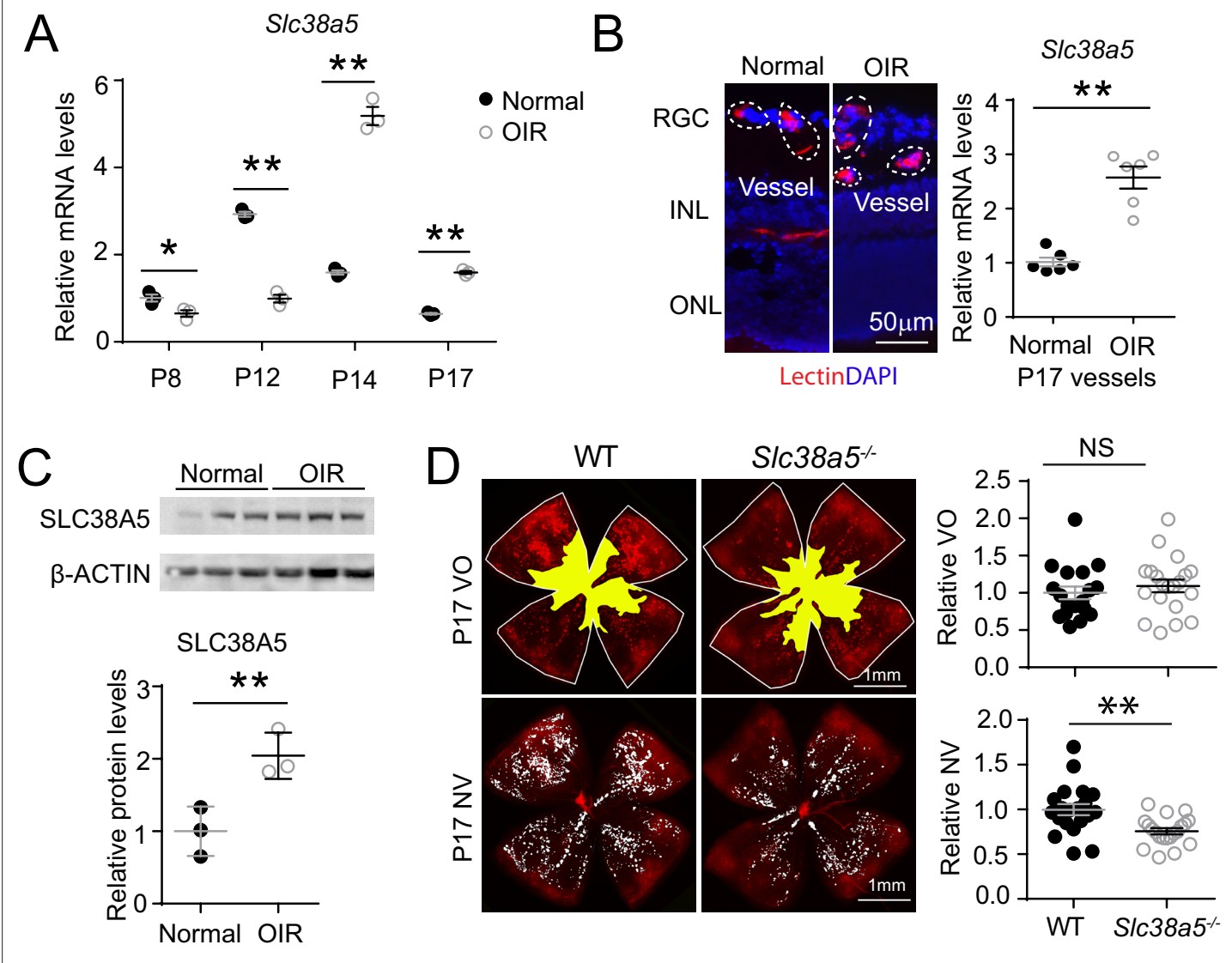

**Figure 4.** *Slc38a5* is enriched in oxygen-induced retinopathy (OIR) pathological neovessels and its deficiency suppresses pathological angiogenesis in OIR. (**A**) *Slc38a5* mRNA expression was measured by RT-qPCR at P8, P12, P14, and P17 in C57BL/6J OIR retinas compared with age-matched normoxic control mice. *Slc38a5* mRNA levels were decreased during hyperoxia stage (P8 and P12) and increased in hypoxia stage (P14 and P17). (**B**) *Slc38a5* mRNA expression was analyzed using RT-qPCR in laser capture micro-dissected pathological neovessels from P17 unfixed C57BL/6J OIR retinas compared with normal vessels isolated from P17 normoxic retinas. Images on the left are representative retinal cross-sections from normal and OIR retinas stained with isolectin B₄ (red) and DAPI (blue), with dotted lines highlighting micro-dissected retinal vessels. GCL: ganglion cell layer, INL: inner nuclear layer, ONL: outer nuclear layer. (**C**) Protein levels of SLC38A5 (52 kDa) were increased in C57BL/6J OIR retinas at P17 compared with normoxic controls using Western blot and quantified with densitometry. Proteins were normalized to β-ACTIN (42 kDa). (**D**) *Slc38a5*⁻/⁻ exposed to OIR had decreased levels of pathological NV (neovascularization) compared with WT OIR controls bred in the same colony at P17. There was no significant difference in VO (vaso-obliteration) between the two groups. Scale bar: 50 µm (**B**), 1 mm (**D**). Each dot represents one retina. Data are expressed as mean ± SEM. n=3–6 per group (**A–C**), n=20 per group (**D**). *p≤0.05; **p≤0.01; n.s.: not significant.

The online version of this article includes the following source data and figure supplement(s) for figure 4:

**Source data 1.** Raw data for *Figure 4*.

**Figure supplement 1.** Quantification of vaso-obliteration of P12 wild type (WT) and *Slc38a5*⁻/⁻ retinas in oxygen-induced retinopathy shows comparable levels.

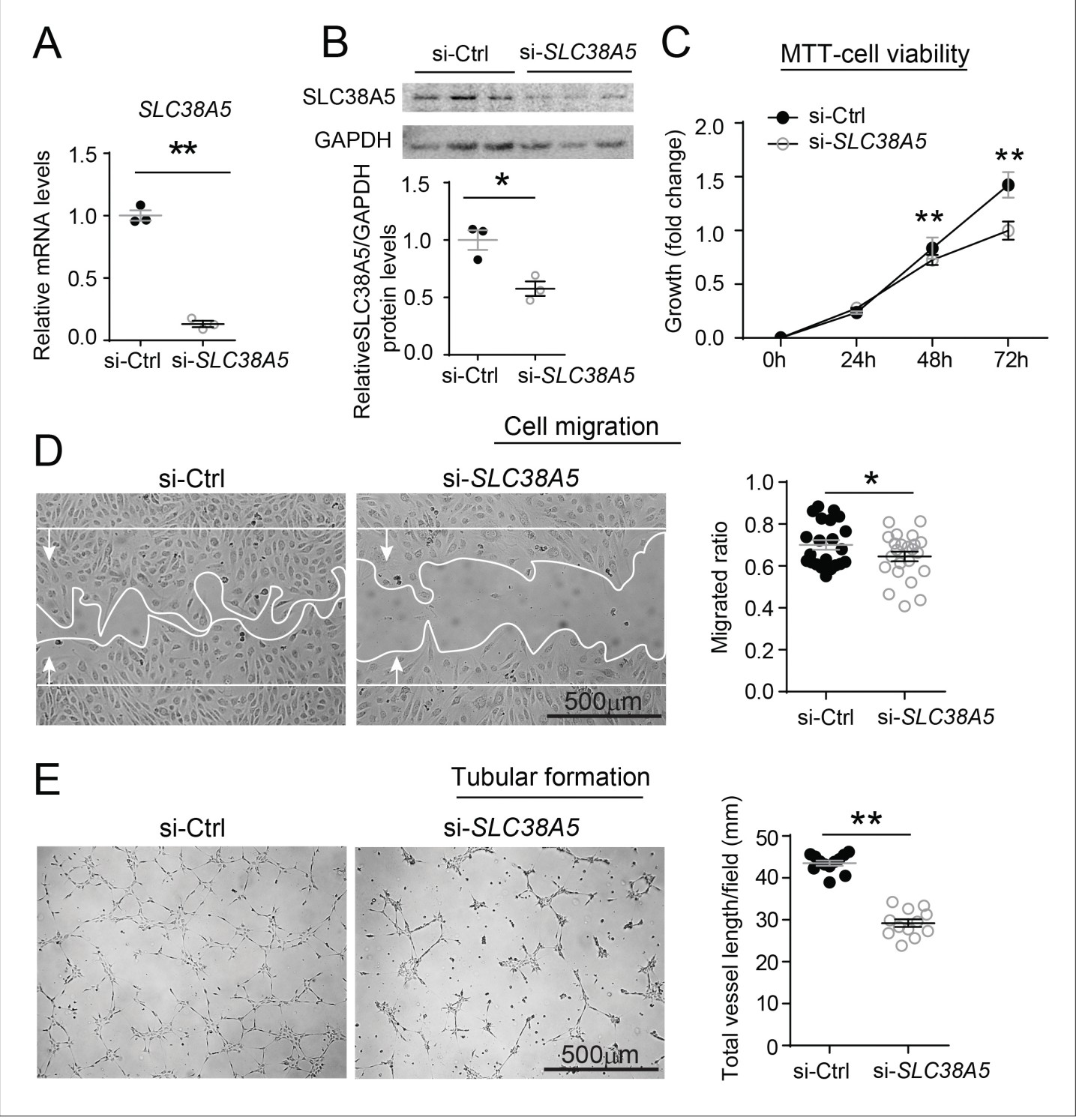

**Figure 5.** Inhibition of *SLC38A5* dampens endothelial cell viability, migration, and tubular formation in vitro. Human retinal microvascular endothelial cells (HRMECs) were transfected with siRNA targeting *SLC38A5* (si- *SLC38A5*) or control siRNA (si-Ctrl). (**A–B**) mRNA (**A**) and protein (**B**) levels of *SLC38A5* (52 kDa) confirm successful knock down by si-*SLC38A5*. Protein levels were normalized to GAPDH (37 kDa). (**C**) HRMEC cell viability was measured with MTT assay. Cell growth rate was calculated as fold change normalized to the values at 0 hr. (**D**) HRMECs were grown to confluence and treated with si-*SLC38A5* or si-Ctrl for 48 hr. Cells were then treated with mitomycin to stop cell proliferation. A scratch was performed to the cells to generate a wound. Migrated areas (new cell growth areas normalized by original wound areas) of HRMECs were measured after 16 hr. (**E**) Tubular formation assay was conducted by collecting cells after 48 hr of si-*SLC38A5* transfection and seeding cells onto Matrigel-coated wells to grow for

*Figure 5 continued on next page*

*Figure 5 continued*

additional 9 hr. Representative images show formation of endothelial cell tubular network, and total vessel length per field was analyzed by Image J. Scale bar: 500 µm (**D&E**). Data are shown as mean ± SEM; n=3–6/group. *p≤0.05; **p≤0.01.

The online version of this article includes the following source data for figure 5:

**Source data 1.** Raw data for *Figure 5*.

## Suppression of SLC38A5 alters key pro-angiogenic factor receptor and signaling in ECs

To understand the factors mediating the effect of SLC38A5 on EC angiogenesis, we evaluated expression of multiple angiogenic factor receptors in response to SLC38A5 knockdown. Suppression of SLC38A5 led to substantially altered mRNA expression of VEGF receptors (VEGFR1 and VEGFR2), Tie2, FGF receptors (FGFR1-3), IGF1R, and IGF2R (*Figure 7A*). In addition, there was substantial suppression of expression of ERK1 and 2 and mTOR (*Figure 7A*), key mediators downstream of VEGFR2 signaling that controls EC growth. As VEGF-A is one of main angiogenic growth factors, we measured the protein levels of VEGF receptors including VEGFR2, the major angiogenic receptor for VEGF-A, and VEGFR1, often acting as a VEGF decoy trap to modulate VEGFR2 function (*Rahimi, 2006*). Protein levels of VEGFR1 were substantially up-regulated, whereas VEGFR2 levels were substantially suppressed after SLC38A5 knockdown in HRMECs (*Figure 7B*). These findings suggest that suppression of SLC38A5 dampens EC glutamine uptake, and subsequently, leads to altered expression of growth factor receptors such as VEGFR1 and 2 to restrict EC angiogenesis.

## Discussion

This study establishes that AA transporter SLC38A5 is a new pro-angiogenic regulator in the retinal vascular endothelium both during development and in retinopathy. SLC38A5 transcription is regulated by Wnt signaling, a fundamentally important pathway in retinal angiogenesis (*Ye et al., 2010*; *Wang et al., 2019a*). We demonstrate that SLC38A5-deficient retinas have delayed developmental angiogenesis and dampened pathological neovascularization in an OIR. Based on these findings, we suggest that SLC38A5 regulates retinal angiogenesis through modulating vascular EC uptake of AAs such as glutamine and thereby influencing angiogenic receptor signaling including VEGFR2 to impact EC growth and function (*Figure 8*).

Previous studies identified SLC38A5 localization in brain glial cells (*Cubelos et al., 2005*) and in the retinal Müller glia and retinal ganglion cells (*Umapathy et al., 2005*; *Umapathy et al., 2008*), yet studies on its function in the brain and eyes are scarce, other than work on its intrinsic role as a glutamine transporter to provide precursor for the neurotransmitter glutamate (*Hamdani et al., 2012*; *Rodríguez et al., 2014*). Elsewhere in the body, identification of SLC38A5 as a marker of pancreatic alpha cell precursor expanded its role in regulating alpha cell proliferation and hyperplasia through nutrient sensing (*Stanescu et al., 2017*; *Dean et al., 2017*; *Kim et al., 2017b*). In the intestine, SLC38A5 was found in crypt cells and may participate in chronic intestinal inflammation (*Singh et al., 2018*; *Singh et al., 2015*). Here, our findings with LCM and single-cell transcriptome analysis identified specific enrichment of *Slc38a5* in retinal vascular endothelium, thus uncovering its new role as an angiogenic regulator and a new marker of sprouting neovessels.

Upregulation of *Slc38a5* in retinal neovessels is potentially regulated by the Wnt/β-catenin signaling pathway, as indicated by our findings in *Lrp5*$^{-/-}$ and *Ndp*$^{y/-}$ mice and in EC cell culture. We showed that *Slc38a5* is a new downstream target gene of Wnt/β-catenin signaling, and that it is deficient in both *Lrp5*$^{-/-}$ and *Ndp*$^{y/-}$ retinas and blood vessels. Our prior work found that Wnt signaling is enriched in developing and pathological neovessels in OIR (*Chen et al., 2011*), further supporting the notion that the Wnt pathway induces *Slc38a5* transcription in neovessels to promote angiogenesis. Delayed development of *Slc38a5*$^{-/-}$ retinal vessels, however, is resolved by adult age with normal vasculature and intact deep layer of vessels (*Figure 3—figure supplements 1 and 2*), indicating that *Slc38a5*$^{-/-}$ retinas do not reproduce FEVR-like symptoms as seen in *Lrp5*$^{-/-}$ and *Ndp*$^{y/-}$ mice. This suggests that down-regulation of *Slc38a5* may only partially explain the effects of Wnt signaling on angiogenesis, and other Wnt- and β-catenin-mediated factors are still at work to drive defective angiogenesis in *Lrp5*$^{-/-}$ and *Ndp*$^{y/-}$ retinas, including, for example, Sox family proteins (*Ye et al.,*

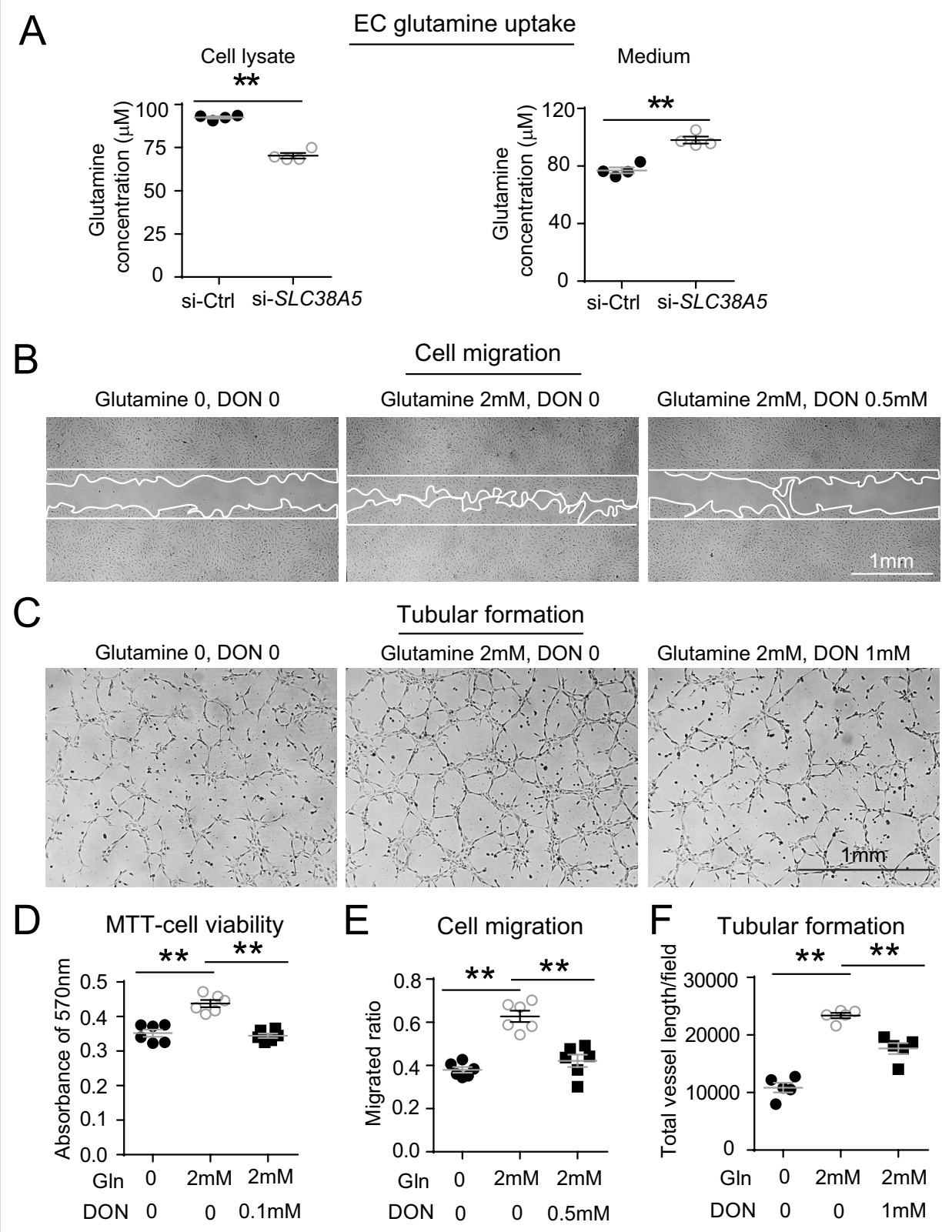

**Figure 6.** SLC38A5 facilitates endothelial cell (EC) uptake of glutamine, which is essential for EC viability, migration, and tubular formation. (**A**) SLC38A5 knockdown with si-*SLC38A5* suppressed glutamine uptake by human retinal microvascular endothelial cells (HRMECs), with decreased glutamine levels in HRMEC cell lysates and increased culture medium levels, measured with a glutamine/glutamate-Glo bioluminescent assay. Levels of glutamine/ glutamate in HRMECs and culture medium samples were determined from bioluminescence readings by comparison to a standard titration curve. (**B**

*Figure 6 continued on next page*

*Figure 6 continued*

**& E**) HRMECs were grown to confluence, and a scratch was applied to generate a wound. Mitomycin was used to stop cell proliferation. A glutamine antagonist, 6-diazo-5-oxo-norleucine (DON), was used to broadly inhibit glutamine uptake. 16 hr were given to the cells to migrate. Representative images are shown in (**B**), and the quantification of migrated areas is shown in (**E**). (**C, F**) HRMECs treated were seeded onto Matrigel for 9 hr and treated with glutamine and DON for tubular formation. Representative images are shown in (**C**), and the quantification of total vessel length per field is shown in (**F**). (**D**) HRMEC cell viability was measured at 24 hr by MTT assay and normalized to the levels at 0 hr to quantify the cell growth rate. Scale bars: 1 mm (**B&C**). Data are expressed as means ± SEM. n=4–6 per group. *p≤0.05; **p≤0.01.

The online version of this article includes the following source data for figure 6:

**Source data 1.** Raw data for *Figure 6*.

2009) and integrin-linked kinase (*Park et al., 2019*). In addition to regulating retinal vessel growth, Wnt/β-catenin signaling is critical to maintain blood vessel barrier and prevent vascular leakage in the brain and eyes (*Ye et al., 2010*; *Wang et al., 2019a*; *Ben and Liebner, 2022*; *Wang et al., 2019b*). *Slc38a5$^{-/-}$* eyes exhibit normal blood-retinal barrier with no detectable vascular leakage in fundus

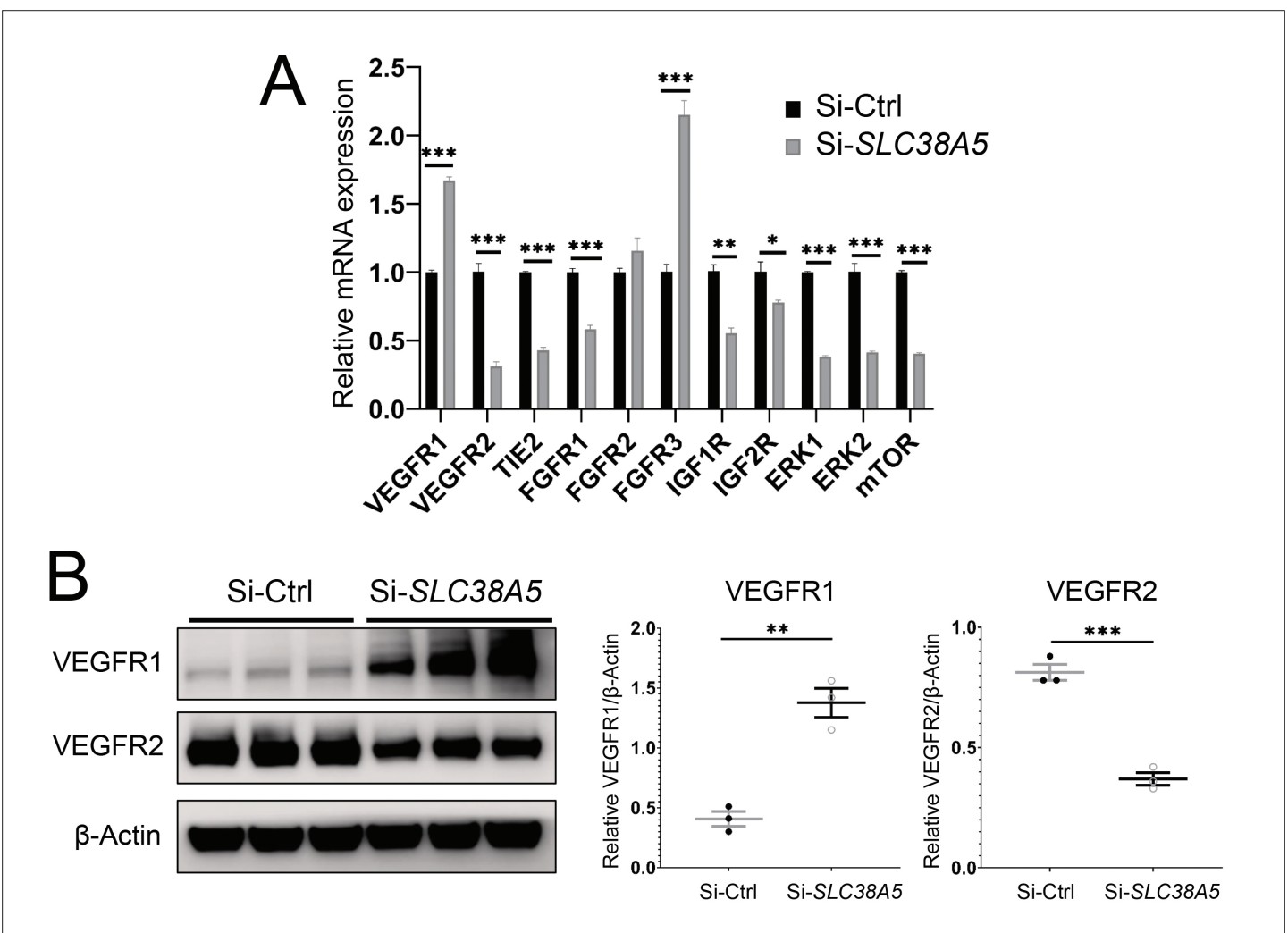

**Figure 7.** Suppression of Slc38a5 modulates growth factor receptors including vascular endothelial growth factor receptor 1 (VEGFR1) and VEGFR2. Human retinal microvascular endothelial cells were transfected with siRNA targeting *SLC38A5* (si-*SLC38A5*) or control siRNA (si-Ctrl) for 72 hr and collected for RT-qPCR or Western blots. (**A**) mRNA levels of growth factor receptors and signaling molecules were normalized by expression of 18 S (n=3–6/group). (**B**) Western Blots show protein levels of VEGFR1 and VEGFR2 with Si-*SLC38A5* or si-Ctrl treatment. Data are shown as mean ± SEM; n=3/group. **p≤0.01; ***p≤0.001.

The online version of this article includes the following source data for figure 7:

**Source data 1.** Raw data for *Figure 7*.

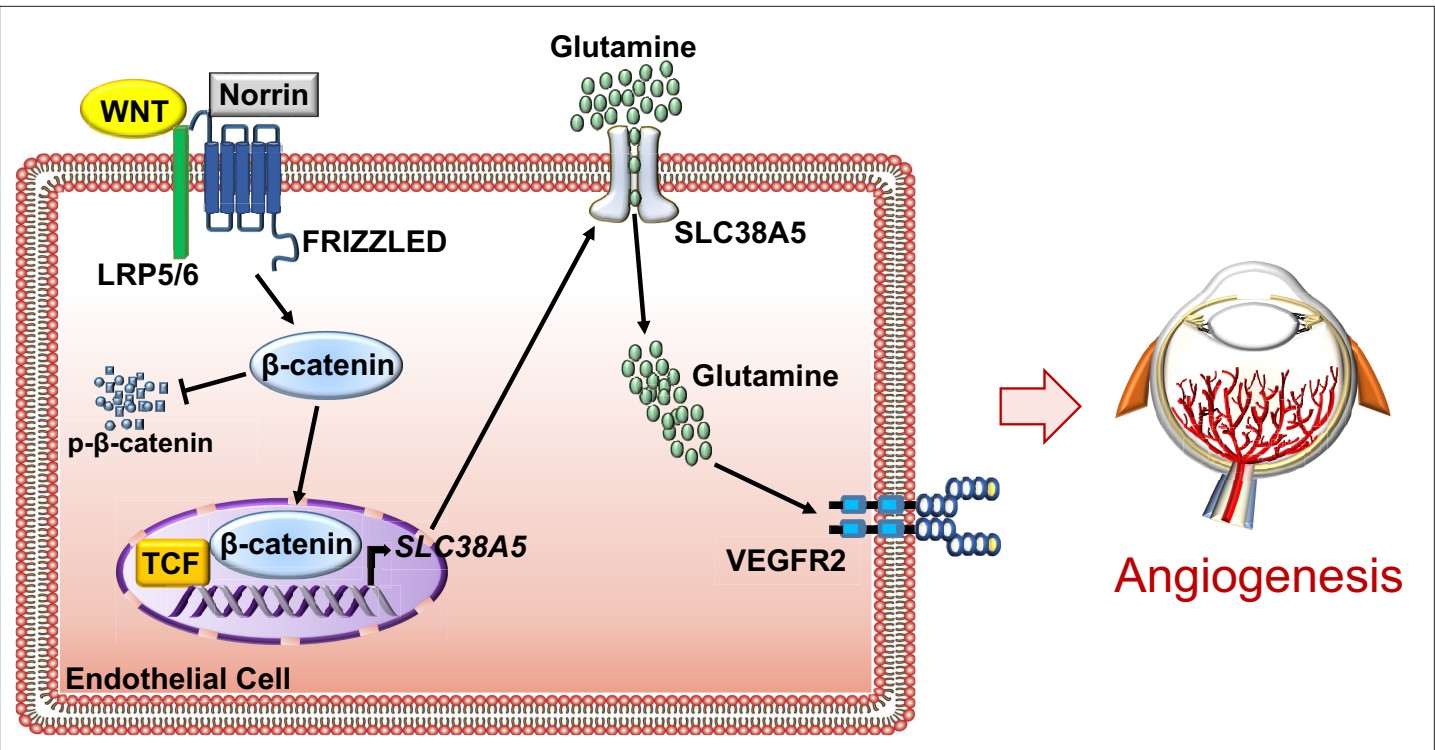

**Figure 8.** Schematic illustration of a pro-angiogenic role of amino acid (AA) transporter SLC38A5 in retinal angiogenesis. In vascular endothelial cells (ECs), Wnt ligands (Wnts and Norrin) activate Wnt/β-catenin signaling, which controls the transcription of EC-enriched SLC38A5 by potentially binding to a TCF-binding site on SLC38A5 promoter or through an indirect transcriptional mechanism. Endothelial SLC38A5 facilitates EC uptake of AAs such as glutamine as energy fuel and source of protein synthesis. Altered glutamine and nutrient availability in EC subsequent affects VEGFR2 levels and signaling, and thus retinal angiogenesis. In retinopathy, expression of both Wnt receptors and endothelial SLC38A5 is enriched in pathological neovessels, promoting glutamine availability and thereby contributing to regulation of VEGFR2 mRNA transcription and protein signaling, resulting in formation of pathologic retinal neovascularization. Inhibition of SLC38A5 may suppress pathologic neovessels and alleviate pathologic neovascularization in retinopathy.

fluorescent imaging (*Figure 3—figure supplement 1*), suggesting that SLC38A5 is dispensable in mediating the effects of Wnt signaling on the retinal vascular barrier largely through both tight junctions such as claudin 5 (*Chen et al., 2012*; *Chen et al., 2011*) and regulation of EC transcytosis (*Wang et al., 2020*). Other than Wnt signaling, *Slc38a5* may potentially be responsive to oxygen sensing and regulated by hypoxia/HIF (hypoxia-inducible factor), as *Slc38a5* expression is suppressed in the phase I of OIR during oxygen exposure and upregulated in the second phase of relative hypoxia. Therefore Wnt signaling may also regulate *Slc38a5* transcription indirectly through a secondary mechanism such as hypoxia through crosstalk with HIF pathway. One might note that mRNA levels of *Slc38a5* showed more drastic suppression than protein levels in Wnt deficient retina and with siRNA inhibition, potentially because of poor correlation between mRNA and protein levels due to their differential regulatory mechanisms.

EC metabolism has been increasingly recognized to play important roles in driving angiogenesis in vascular development and maturation (*Draoui et al., 2017*; *Teuwen et al., 2019*). Specifically, AAs including glutamine and asparagine can drive EC proliferation and vessel sprouting through regulation of TCA cycle and protein synthesis (*Huang et al., 2017*; *Kim et al., 2017a*), since either depletion of glutamine or inhibition of glutaminase 1 (an enzyme that converts glutamine to glutamate), or suppression of asparagine synthetase, impairs angiogenesis (*Huang et al., 2017*; *Kim et al., 2017a*). Moreover, glutamine synthetase (another enzyme responsible for glutamine formation) promotes angiogenesis through activating small GTPase RHOJ (ras homolog family member J), independent of its role in glutamine synthesis (*Eelen et al., 2018*). Our data showed that glutamine uptake in vascular endothelium is substantially down-regulated when SLC38A5 is suppressed, suggesting that SLC38A5 may regulate glutamine availability to ECs and thus control EC angiogenesis and metabolism. Our findings on SLC38A5 and its transport of glutamine in EC, together with

the effect of glutamine on EC angiogenic function, further strengthen the idea that glutamine is key to sprouting angiogenesis and suggest its bioavailability in EC is controlled in part by SLC38A5. Glutamine was reported previously to fuel EC proliferation but not migration in HUVEC (human umbilical vein endothelial cells) in vitro (*Kim et al., 2017a*), yet in our study we found glutamine promotes migration of HRMECs, which may reflect different cell-specific responses to glutamine in macrovascular HUVECs vs. microvascular HRMECs, as well as different migration assay methods used.

While glutamine is the most abundant AA in circulation, Müller glia also provide glutamine locally in the retina through uptake of and synthesis from excess extracellular glutamate as neurotransmitter (*Bringmann et al., 2013*). Thus SLC38A5 may help vascular ECs to uptake and utilize free glutamine from both systemic circulation and those released by Müller glia to influence AA uptake and metabolism. A recent study found that hyperoxia promotes glutamine consumption and glutamine-fueled anaplerosis in Müller glia (*Singh et al., 2020*), suggesting a link between oxygen sensing and glutamine metabolism in the Müller glia. Whether SLC38A5 may interlink the oxygen-sensing and glutamine metabolism in Müller cells and ECs under other similar physiological or pathological conditions will await further studies.

In addition to glutamine, SLC38A5 may transport other AAs such as serine and glycine, which may further regulate angiogenesis in eye diseases. For example, serine deprivation in the absence of a synthesizing enzyme phosphoglycerate dehydrogenase causes lethal vascular defects in mice (*Vandekeere et al., 2018*), whereas serine-glycine metabolism is also required in angiogenesis and linked to endothelial dysfunction in response to oxidized phospholipids (*Hitzel et al., 2018*). In the mouse model of OIR, serine and one-carbon metabolism were found to mediate HIF response in retinopathy through liver-eye serine crosstalk (*Singh et al., 2019*). In macular telangiectasia type 2 (MacTel), a rare degenerative eye disease with abnormal intraretinal angiogenesis, defective serine biosynthesis, and PHDGH haploinsufficiency was genetically linked with disease onset and associated with toxic ceramide accumulation in circulation and in retinal pigment epithelium cells (*Eade et al., 2021*; *Gantner et al., 2019*). Whether SLC38A5 may regulate serine transportation to influence retinal AA metabolism, angiogenesis, and retinal diseases will need additional investigation.

We showed here that decreased glutamine uptake in EC is directly associated with altered expression of growth factor receptors and particularly increased levels of VEGFR1 and decreased levels of VEGFR2. As a major angiogenic growth factor, VEGF-A exerts its effects on ECs mainly through its receptor VEGFR2 to promote angiogenesis – including EC proliferation, migration, and cellular differentiation (*Abhinand et al., 2016*), whereas VEGFR1 often has negative angiogenic function and acts largely as a decoy receptor of VEGF to limit VEGFR2 function (*Rahimi, 2006*). Suppression of SLC38A5 may thus not only directly decrease VEGFR2 expression but also limit VEGFR2 function due to increased VEGFR1 levels, both of which can result in decreased retinal angiogenesis.

Findings presented here suggest that modulation of SLC38A5 and its associated AAs may have translational value in treating retinopathy. Premature infants often lack conditionally essential AAs such as glutamine or arginine (*Neu, 2003*; *Wu et al., 2004*) due to their impaired endogenous synthesis. Providing much needed AAs early on might promote normalization of delayed vessel growth, and thereby preventing the neovascular phase of retinopathy. Previously, supplement of an arginine-glutamine (Arg-Glu) dipeptide was found to dampen pathological neovascularization in OIR by promoting normal vascular restoration (*Shaw et al., 2018*; *Neu et al., 2006*). Here our data suggest lack of SLC38A5, and likely subsequent impaired glutamine uptake, dampens developmental angiogenesis, in line with the pro-angiogenic role of AAs in the first phase of retinopathy. In the late proliferative phase of retinopathy, however, inhibiting AA transporters like SLC38A5 or starving retinal vessels from AA nutrients may be beneficial in directly suppressing uncontrolled pathological neovascularization. Targeting of AAs and their transporters such as SLC38A5 may thus represent a new potential approach to treat retinopathy via modulating EC AA transport and metabolism.

One limitation of the current study lies in the mutant mice used, which have systemic knockout of SLC38A5. Potential systemic influence of SLC38A5 knockout on circulating factors can be a compounding factor in interpretation of the results. Yet our data from ocular siRNA delivery strongly suggest that local inhibition of SLC38A5 did directly impair retinal angiogenesis, and systemic influence from other organs is likely minimal. It is also not clear whether other retinal cell types, including glia and neurons may be impacted by SLC38A5 modulation and thus affect vascular endothelium

metabolism and angiogenesis, although our cell culture data largely supports an EC-specific pro-angiogenic role of SLC38A5 via regulating cellular AA availability and metabolism.

In summary, our data present direct evidence that SLC38A5 is a novel regulator of EC AA metabolism and retinal angiogenesis. Expression of SLC38A5 is enriched in sprouting neovessels and driven by Wnt/β-catenin signaling pathway, as evident in Wnt-deficient retinas. Suppression of SLC38A5 may limit glutamine uptake by ECs, resulting in altered cellular metabolism, dampened VEGFR2 signaling and blunted retinal angiogenesis. Our findings of SLC38A5 as a modulator of pathologic retinal angiogenesis suggest the possibility of targeting SLC38A5 and its transported AAs as new therapeutic intervention for the treatment of vascular eye diseases and potentially other angiogenesis-related diseases.

# Materials and methods

## Key resources table

| Reagent type (species) or resource | Designation | Source or reference | Identifiers | Additional information |
|---|---|---|---|---|
| Genetic reagent (*Mus musculus*) | *Slc38a5*$^{-/-}$ | J.Kim et al., Cell Metab **25**, (2017). PMID:28591637 | | |
| Genetic reagent (*M. musculus*) | *Lrp5*$^{-/-}$ | Jackson Laboratory | Stock no. 005823 | |
| Genetic reagent (*M. musculus*) | *Ndp*$^{y/-}$ | Jackson Laboratory | Stock no. 012287 | |
| Genetic reagent (*M. musculus*) | C57BL/6J | Jackson Laboratory | Stock no: 000664 | |
| Antibody | SLC38A5 (Rabbit polyclonal) | Biorbyt | orb317962 | Dilution Western: 1:1000 IHC: 1:200 |
| Antibody | Non-phosphorylated β-catenin (Rabbit polyclonal) | Cell Signaling Technology | 8814 S | Dilution 1:1000 |
| Antibody | β-Catenin (Rabbit polyclonal) | Santa Cruz Biotechnology | sc-7199 | Dilution 1:1000 |
| Antibody | Glyceraldehyde-3-phosphate dehydrogenase (GAPDH) (Mouse monoclonal) | Santa Cruz Biotechnology | sc-32233 | Dilution 1:2000 |
| Antibody | β-ACTIN (Mouse monoclonal) | Sigma-Aldrich | A1978 | Dilution 1:1000 |
| Antibody | VEGFR1 (Rabbit polyclonal) | Cell Signaling Technology | 2893 S | Dilution 1:2000 |
| Antibody | VEGFR2 (Rabbit monoclonal) | Cell Signaling Technology | 2479 | Dilution 1:2000 |
| Antibody | Anti-mouse IgG, HRP-conjugated (Sheep polyclonal) | Sigma-Aldrich | NA9310 | Dilution 1:10000 |
| Antibody | Anti-rabbit IgG, HRP-conjugated (Donkey polyclonal) | Sigma-Aldrich | SAB3700934 | Dilution 1:10000 |

## Animals

All animal studies described in this paper were approved (protocol no. 00001722) by the Boston Children's Hospital Institutional Animal Care and Use Committee (IACUC) and also adhered to the ARVO Statement for the Use of Animals in Ophthalmic and Vision Research. *Slc38a5*$^{-/-}$ mice (100% of C57BL/6 background) were generated and knockout validated previously (*Kim et al., 2017b*). Both male and female *Slc38a5* knockout mice were used in experiments, and littermate WT controls from the same breeding colony were used as comparison. C57BL/6J mice were obtained from Jackson Laboratory (stock no: 000664) and were used for siRNA treatment and LCM experiments as well as WT control mice for *Lrp5*$^{-/-}$ (stock no. 005823). *Ndp*$^{y/-}$ (stock no. 012287) were also obtained from the Jackson Laboratory (Bar Harbor, ME). Male *Ndp*$^{y/+}$ mice were used as control for *Ndp*$^{y/-}$ for X-linked *Ndp* gene.

## Oxygen-induced retinopathy

The OIR mouse model was performed as previously described (*Smith et al., 1994*; *Liu et al., 2017*). Newborn mouse pups with nursing mothers were exposed to 75 ± 2% oxygen from P7 to P12 and returned to room air until P17. Mice were anesthetized with ketamine/xylazine and euthanized by cervical dislocation, followed by retinal dissection and blood vessel staining.

## Intravitreal injection of siRNA

Intravitreal injections were performed in C57BL/6J mouse pups at various developmental time points or with OIR, according to previously established protocols (*Chen et al., 2009*; *Liu et al., 2015*; *Liu et al., 2019*; *Sun et al., 2017*). Briefly, mice were anesthetized with isoflurane in oxygen. 1 µg of *si*-Slc38a5 (ThermoFisher, Cat no. 4390771) dissolved in 0.5 µL of vehicle solution was injected using a 33-gauge needle behind the limbus of the eye, whereas the contralateral eye of the same animal was injected with an equal amount of negative control scrambled siRNA (si-Ctrl) (ThermoFisher, Cat no. 4390844). After injection, eyes were lubricated with sterile saline, and an antibiotic eye ointment was applied. At specified days after injection, mice were sacrificed, and retinal vasculature was analyzed.

## Retinal dissection and vessel staining

Mouse pups at various developmental time points or post-OIR exposure were sacrificed, retinas dissected, and blood vessels stained according to previous protocols (*Liu et al., 2017*). Isolated eyes were fixed in 4% paraformaldehyde in phosphate-buffered saline (PBS) for 1 hr at room temperature. Retinas were dissected, stained with labeled *Griffonia simplicifolia* Isolectin $B_4$ (Alexa Fluor 594 conjugated, Cat no. 121413; Invitrogen), and flat-mounted onto microscope slides (Superfrost/Plus, 12-550-15; Thermo Fisher Scientific, Waltham, MA) with photoreceptor side down and embedded in antifade reagent (SlowFade, S2828; Invitrogen). Retinas were imaged with a fluorescence microscope (AxioObserver.Z1 microscope; Carl Zeiss Microscopy), and images were merged to cover the whole flat-mounted retina. For imaging deeper retinal layers, fixed retinas were permeabilized first with PBS in 1% Triton X-100 for 30 min, followed by isolectin $B_4$ staining with 0.1% TritonX-100 in similar processes as described above.

## Quantification of retinal vascular development, vaso-obliteration, and pathological neovascularization in OIR

Quantification of developmental vasculature was performed by using Adobe Photoshop (Adobe Systems, San Jose, CA, USA) and ImageJ from NIH according to previous protocols (*Stahl et al., 2010*; *Liu et al., 2017*). Retinal vascular areas were expressed as a percentage of the total retinal areas. n is the number of retinas quantified.

Quantification of retinal vaso-obliteration and neovascularization in OIR retinas followed previously described protocols (*Stahl et al., 2009*; *Connor et al., 2009*; *Banin et al., 2006*) by using Adobe Photoshop and ImageJ. The avascular area absent of isolectin $B_4$ staining was outlined and calculated as a percentage of the whole retinal area. Pathological neovascularization was recognized by the abnormal aggregated morphology and was quantified using a computer-aided SWIFT_NV method and also normalized as percentage of the whole retina area (*Stahl et al., 2009*). Quantification was done with the identity of the samples masked.

## Laser capture microdissection of retinal vessels

LCM of retinal vessels was carried out based on previous protocols (*Wang et al., 2020*; *Liu et al., 2015*; *Li et al., 2014*). Mouse eyes were embedded in optimal cutting temperature compound, sectioned at 10 µm, and mounted on polyethylene naphthalate glass slides (Cat no. 115005189, Leica). Frozen sections were dehydrated, briefly washed, then stained with isolectin $B_4$ to visualize blood vessels. Retinal blood vessels were laser capture microdissected with Leica LMD 6000 system (Leica Microsystems). Micro-dissected samples were collected in lysis buffer from the RNeasy micro kit (Cat no. 74004, Qiagen, Chatsworth, MD, USA), followed by RNA isolation and RT-qPCR.

## Single-cell transcriptome analysis

Gene expression of *Slc38a5* and EC marker *Pecam1* in mouse retinal cells types was identified using the online single-cell dataset: Study - P14 C57BL/6J mouse retinas (https://singlecell.broadinstitute.

org/single_cell/study/SCP301) (*Macosko et al., 2015*). Similarly, gene expression of *SLC38A5* and *PECAM1* was identified in human retinal single-cell dataset - Cell atlas of the human fovea and peripheral retina (https://singlecell.broadinstitute.org/single_cell/study/SCP839) (*Yan et al., 2020*). Both studies are accessed from Single Cell Portal, Broad Institute. Dot plots of gene expression for different retinal cell types were grouped and displayed.

## EC cell culture and assays of angiogenic function and glutamine uptake

HRMECs (Cat no. ACBRI 181, Cell system) were authenticated by their characteristic EC morphology and positive expression of EC markers, and negative of mycoplasma contamination. Cells were cultured in completed endothelial culture medium supplemented with culture boost-R (*Wang et al., 2020*). Cells between passages 4 and 7 were transfected with si-*SLC38A5* siRNA (Cat no. 4392420, Thermo Fisher Scientific) or negative control siRNA (si-Ctrl, Cat no. AM4611, Thermo Fisher Scientific). siRNA knockdown was confirmed by RT-qPCR and Western blot of cells collected 48–72 hr after transfection.

HRMEC viability and/or proliferation was assessed at 48 and 72 hr after transfection with *si-SLC38A5* or si-Ctrl using an MTT cell metabolic activity assay kit (Cat no. V13154, Life Technologies) as described previously (*Liu et al., 2019*). Briefly, HRMECs were incubated for 4 hr in solutions containing a yellow tetrazolium salt MTT (3-(4,5-dimethylthiazol-2-yl)–2,5-diphenyltetrazolium bromide). NAD(P)H-dependent oxidoreductase in metabolically active live HRMECs reduces MTT to purple formazan crystals. Afterward a solution containing SDS-HCL was added which dissolves the purple formazan, followed by measurement of absorbance at 570 nm using a multi-well spectrophotometer. Measurement of MTT cellular metabolic activity is indicative of cell viability, proliferation, and cytotoxicity. HRMEC migration and tube formation assays were carried out according to previous protocols (*Liu et al., 2015*).

Glutamine uptake by HRMEC was performed using a glutamine/glutamate-Glo bioluminescent assay (Progema, Cat no. J8021) according to manufacture protocols. HRMEC cells were treated with si-*SLC38A5* or si-Ctrl at passage 6. 48 hr after siRNA transfection, culture medium of each well was replaced with fresh culture medium at equal volume. 72 hr post siRNA treatment, both culture medium and cells were harvested for the glutamine/glutamate-Glo assay. Amounts of glutamine/glutamate in samples were determined from luminescence readings by comparison to a standard titration curve.

## SLC38A5 promoter cloning and dual-luciferase reporter assay

Cloning and reporter assays were performed based on previous protocols (*Wang et al., 2020*). Putative TCF-binding motif (A/TA/TCAAAG) was identified in three mouse *Slc38a5* promoter regions, which were amplified by PCR using the following primers: *Slc38a5*_P1, F: 5'-TATCGCTAGCCCAGCAGGGT GTATTTATG-3' and R: 5'-TATCCTCGAGGGGAGCGCTTTCAATCCTCAG-3'; *Slc38a5*_P2, F: 5'-TATC GCTAGC TCTCAAACTGTATCATGGAG-3' and R: 5'- TATCCTCGAGCTACTTGCTGAAGACGTTG-3'; *Slc38a5*_P3, F: 5'-TATCGCTAGCAGGTCCTCTGAAGTATTGATC-3' and R: 5'-TATCCTCGAGAGGGAGA GTTCAAGTGTAGGT-3'. PCR products were purified by gel extraction, cloned into the pGL3 promoter luciferase vector (Promega, Madison, WI; E1751), and verified by Eton Bioscience (Boston, MA) with 100% matching sequence matching to the promoter region of *Slc38a5*. Both the promoter plasmids and a stabilized active form of β-catenin plasmid were transfected into HEK293T cells. After 48 hr, luciferase activity was measured with a dual-luciferase reporter assay kit (Promega; E1910), and the relative luciferase activity was determined by normalizing the firefly luciferase activity to the respective *Renilla* luciferase activity. At least three rounds of experiments were performed for the luciferase assays, and for each experiment each transfection group has at least five replicates.

## RNA isolation and real-time RT-PCR

Total RNA was isolated from mouse retinas or from HRMEC culture with RNeasy Kit (Qiagen) based on manufacturer protocols. For LCM isolated vessels, RNA was isolated with the RNeasy Micro Kit (Qiagen, Cat no. 74004). Synthesis of cDNA and q-PCR was performed using established standard protocols.

Primers for mouse real-time RT-PCR include: *Slc38a5*: F:5'- GACCTTTGGATACCTCACCTTC-3' and R: 5'- CCAGACGCACACAAAGGATA-3'; *Rn18s*: F: 5'- CACGGACAGGATTGACAGATT-3' and R: 5'- GCCAGAGTCTCGTTCGTTATC-3'.

Human primers include: *SLC38A5*: F:5'- GAAGGGAAACCTCCTCATCATC-3' and R: 5'- CAGG TAGCCCAAGTGTTTCA-3'; *18 S:* F: 5'- GCCTCGAAAGAGTCCTGTATTG-3' and R: 5'- TGAAGAGG GAGCCTGAGAAA-3'; *FLT1*(VEGFR1): F: 5'- CCGGCTCTCTATGAAAGTGAAG-3' and R: 5'- CGAG TAGCCACGAGTCAAATAG-3'; *KDR*(VEGFR2): F: 5'- AGCAGGATGGCAAAGACTAC-3' and R: 5'- TACTTCCTCCTCCTCCATACAG-3'; *TEK*(TIE2): F: 5'- TTTGCCCTCCTGGGTTTATG-3' and R: 5'- CTTG TCCACTGCACCTTTCT-3'; *FGFR1:* F: 5'- GAGGCTACAAGGTCCGTTATG-3' and R: 5'- GATGCTGC CGTACTCATTCT-3'; *FGFR2*: F: 5'- GGATAACAACACGCCTCTCTT-3' and R: 5'- CTTGCCCAGTGT CAGCTTAT –3'; *FGFR3*_F: 5'- CGAGGACAACGTGATGAAGA-3' and R: 5'- TGTAGACTCGGTCAAA CAAGG-3'; *IGF1R*: F:5'- CATGGTGGAGAACGACCATATC-3' and R: 5'- GAGGAGTTCGATGCTG AAAGAA-3'; *IGF2R*: F: 5'- CAGCGGATGAGCGTCATAAA-3' and R: 5'- CGTGTCCCATGTGAAGAAGT AG-3'; *MAPK3*(ERK1): F: 5'- GCTGAACTCCAAGGGCTATAC-3' and R: 5'- GTTGAGCTGATCCAGG TAGTG-3'; *MAPK1*(ERK2): F: 5'- GGTACAGGGCTCCAGAAATTAT-3' and R: 5'- TGGAAAGATGGG CCTGTTAG-3'; *MTOR*: F: 5'- GGGACTACAGGGAGAAGAAGAA-3' and R: 5'- GCATCAGAGTCA AGTGGTCATAG-3'.

## Western blot

Western blot was performed based on standard protocols (*Wang et al., 2020*). Retinal or HRMEC samples were extracted and sonicated in radioimmunoprecipitation assay lysis buffer (Thermo Fisher Scientific; 89901) with protease and phosphatase inhibitors (Sigma-Aldrich; P8465, P2850). Protein concentration was determined with a BCA (bicinchoninic acid) protein assay kit. Equal amounts of protein were loaded in NuPAGE bis-tris protein gels (Thermo Fisher Scientific) and electroblotted to a polyvinylidene difluoride membrane. After blocking with 5% nonfat milk for 1 hr, membranes were incubated with a primary monoclonal antibody overnight at 4°C, followed by washing and incubation with secondary antibodies with horseradish peroxidase-conjugation (Amersham) for 1 hr at room temperature. Chemiluminescence was generated by incubation with enhanced chemiluminescence reagent (Thermo Fisher Scientific; 34075) and signal detected using an imaging system (17001401, Bio-Rad, Hercules, CA). Densitometry was analyzed with ImageJ software. Primary antibodies: anti-SLC38A5 (Biorbyt, St. Louis, MO; orb317962,), anti-non-phosphorylated β-catenin (Cell Signaling Technology; 8814 S), anti–β-catenin (Santa Cruz Biotechnology, Santa Cruz, CA; sc-7199), anti-glyceraldehyde-3-phosphate dehydrogenase (Santa Cruz Biotechnology; sc-32233), anti-VEGFR1 (Cell Signaling Technology; 2893 S), anti-β-actin (Sigma-Aldrich; A1978) and anti-VEGFR2 (Cell Signaling Technology; 2479). HRP-linked secondary antibodies were from Sigma-Aldrich: anti-mouse antibody (NA9310); anti-rabbit antibody (SAB3700934).

## Statistical analysis

Data were analyzed using GraphPad Prism 6.01 (GraphPad Software, San Diego, CA). Results are presented as means ± SEM from animal studies; means ± SD for non-animal studies with at least three independent experiments. Statistical differences between groups were analyzed using a one-way ANOVA statistical test with Dunnett's multiple comparisons tests (more than two groups) or two-tailed unpaired *t* tests (two groups); p<0.05 was considered statistically significant.

## Acknowledgements

We thank Drs. Lois E H Smith, Ye Sun, and Bertan Cakir for very helpful discussion. Funding: This work was supported by NIH/NEI R01 grants (EY028100, EY024963, and EY031765), Boston Children's Hospital Ophthalmology Foundation, and Mass Lions Eye Research Fund Inc (to JC), and Knights Templar Eye Foundation Career Starter Grant (to ZW).

## Additional information

### Funding

| Funder | Grant reference number | Author |
| --- | --- | --- |
| National Eye Institute | EY028100 | Jing Chen |

| Funder | Grant reference number | Author |
|---|---|---|
| National Eye Institute | EY024963 | Jing Chen |
| National Eye Institute | EY031765 | Jing Chen |
| Boston Children's Hospital Ophthalmology Foundation | | Jing Chen |
| Knights Templar Eye Foundation | Career Starter Grant | Zhongxiao Wang |
| Mass Lions Eye Research Fund Inc | | Jing Chen |

The funders had no role in study design, data collection and interpretation, or the decision to submit the work for publication.

## Author contributions

Zhongxiao Wang, Conceptualization, Resources, Data curation, Software, Formal analysis, Funding acquisition, Validation, Investigation, Visualization, Methodology, Writing - original draft, Project administration, Writing – review and editing; Felix Yemanyi, Data curation, Formal analysis, Validation, Investigation, Visualization, Methodology, Writing – review and editing; Alexandra K Blomfield, Shuo Huang, Chi-Hsiu Liu, William R Britton, Steve S Cho, Data curation, Formal analysis, Investigation, Methodology, Writing – review and editing; Kiran Bora, Data curation, Formal analysis, Visualization; Yohei Tomita, Data curation, Investigation, Methodology, Writing – review and editing; Zhongjie Fu, Data curation, Software, Formal analysis, Investigation, Methodology, Writing – review and editing; Jian-xing Ma, Wen-hong Li, Resources, Writing – review and editing; Jing Chen, Conceptualization, Resources, Data curation, Software, Formal analysis, Supervision, Funding acquisition, Validation, Investigation, Visualization, Methodology, Writing - original draft, Project administration, Writing – review and editing

## Author ORCIDs
Zhongjie Fu (iD) http://orcid.org/0000-0002-8182-2983
Jing Chen (iD) http://orcid.org/0000-0002-9183-8698

## Ethics

All animal studies described in this paper were approved (protocol #: 00001722) by the Boston Children's Hospital Institutional Animal Care and Use Committee (IACUC), and also adhered to the ARVO Statement for the Use of Animals in Ophthalmic and Vision Research.

## Decision letter and Author response
Decision letter https://doi.org/10.7554/eLife.73105.sa1
Author response https://doi.org/10.7554/eLife.73105.sa2

# Additional files

## Supplementary files
• Transparent reporting form

## Data availability

All data generated or analyzed during this study are included in the uploaded zip document containing Western blot images and source numerical data for figures 1-7.

The following previously published datasets were used:

| Author(s) | Year | Dataset title | Dataset URL | Database and Identifier |
|---|---|---|---|---|
| Macosko EZ | 2015 | Study - P14 C57BL/6J mouse retinas | https://singlecell.broadinstitute.org/single_cell/study/SCP301 | Single Cell Portal, SCP301 |
| Yan W | 2020 | Study - Cell atlas of the human fovea and peripheral retina | https://singlecell.broadinstitute.org/single_cell/study/SCP839 | Single Cell Portal, SCP839 |

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
