## [Editor Report]

Anti-VEGF treatment is currently used to treat patients with pathological retinal angiogenesis, but finding the underlying cause of increased VEGF has been a challenge for the field. Wang and colleagues determined the role played of amino acid transporter, SLC38A5, in retinal angiogenesis and they showed that SLC38A5 in the retina is under the control of Wnt/β-catenin signaling. The deficiency of SLC38A5 resulted in delayed retinal vascular growth and reduced neovascularization in an Oxygen-Induced Retinopathy model. Additionally, the authors addressed the mechanisms of Slc38a5 as a glutamine transporter regulating retinal vascular development through VEGF receptors.

---

## [Decision Letter]

**Decision letter after peer review:**

Thank you for submitting your article "Amino acid transporter SLC38A5 regulates developmental and pathological retinal angiogenesis" for consideration by *eLife*. Your article has been reviewed by 3 peer reviewers, and the evaluation has been overseen by a Reviewing Editor and Mone Zaidi as the Senior Editor. The following individual involved in review of your submission has agreed to reveal their identity: Yan Gong (Reviewer #3).

Please address all of the concerns of the reviewers. These were seen as Essential Revisions:

1) The conclusion in Figure 1 is that Slc38a5 expression is downregulated in Lrp5 and Ndp knockouts. However, controls used in 1A and B are confusing. Please establish better temporal expression of the transporter in control (WT) retinas. Furthermore, While it's clear that Slc38a5 mRNA and protein expression is enriched in LCM-isolated retinal vessels, it's unclear whether that expression is exclusively in ECs or also in vessel associated mural cells (Figure 1C, Figure S1). Although Figure S1 shows the mining of mouse retinal scRNA-seq database to demonstrate exclusive Slc38a5 expression in ECs, please validate that in the tissue using either RNA in situ hybridization or IHC

2) The conclusion in Figure 2 that Slc38a5 is a direct target of Wnt signaling is based on correlative data, yet a direct interaction (between β-catenin protein and the DNA driving the regulation of Slc38a5 to the retina) is not shown. They are assuming that the 5' flanking regions upstream of Slc38a5 drive retina-specific expression. Another assumption is that the addition of β-catenin is directly binding to these regions. An additional experiment with a construct containing mutant Wnt binding site(s) would be evidence of direct binding.

3) Figure 3 has low quality images with data not matching the images, please address.

4) The conclusion that the blood vessels in the adult are normal (Figure S2) needs to be substantiated with data.

5) A lack of data at P12 in the OIR model is an issue.

6) Cell viability and use of DON with varying concentrations, please address.

7) The authors conclusion that glutamine uptake is essential for cell viability, migration and tubular formation is based on data shown in Figure 6. There is a concern that cell viability is affecting cell migration and tubular formation, as the glutamine inhibitor is used at a much higher concentration for the latter two assays in this figure.

8) Overall, the Discussion needs to emphasize the role of endothelial cell metabolism in vascular development and maturation and how Slc38a5 may influence these processes.

*Reviewer #1 (Recommendations for the authors):*

The study focuses on the role of SLC38A5, a neutral amino acid transporter, in retinal angiogenesis. The authors show that Slc38a5 transporter is highly enriched in normal retinal vascular ECs, and upregulated in the ECs in pathogenic neoangiogenesis (the OIR model). Additionally, the authors show that Slc38a5 transcription is regulated by Wnt/β-catenin signaling and deletion of Slc38a5 in mice substantially delays retinal vascular development and suppresses pathological neovascularization in the OIR model by suppressing glutamine uptake and reducing VEGFR2 expression. The authors claim that SLC38A5 is a new metabolic regulator of retinal angiogenesis.

The study is performed carefully and demonstrates clearly an important role for the transporter in retina angiogenesis. However, there are some concerns that need to be addressed as follows:

1) The authors show that Slc38a5 is downregulated in the Lrp5-/- and Ndpy/- retinas (Figure 1A, B); however, there is a discrepancy in Slc38a5 expression levels in the control retinas. The expression of Slc38a5 in the WT retina goes down from P8-P12 and then plateaus through P17 (Figure 1A). In contrast, in Figure 1B, the expression of Slc38a5 in the Ndpy/+ retina plateaus from P8-P12 and then goes up through P17. The authors need to establish better the temporal expression of the transporter in control (WT) retinas.

2) While it's clear that Slc38a5 mRNA and protein expression is enriched in LCM-isolated retinal vessels, it's unclear whether that expression is exclusively in ECs or also in vessel associated mural cells (Figure 1C, Figure S1). Although Figure S1 shows the mining of mouse retinal scRNA-seq database to demonstrate exclusive Slc38a5 expression in ECs, it's necessary to validate that in the tissue using either RNA in situ hybridization or IHC for in combination with an endothelial cell or mural cell marker.

3) Figure 3: The image qualities are poor. The authors need to enhance image qualities to show the vessels clearly in such low magnification.

4) Figure 3F: The images in this panel show more than 50% decrease in the vascular area in the deep plexus between WT and Slc38a5-/- retinas. However, the graph shows a far lower (10-15% at best) decrease in the vascular coverage. The authors need to select representative images to match the graph.

5) The authors show the presence of vessels in the adult Slc38a5-/- retina to claim that vascular abnormalities seen in early development are gone in the adult (Figure S2). However, the presence of vessels does not mean that there are no vascular abnormalities. The authors should compare established vascular parameters such as branching-density, vascular pruning between adult WT and Slc38a5-/- retinas to justify the claim.

6) While the authors show that there is a decrease in pathological neovascularization in the Slc38a5-/- retina at P17 in the OIR model (Figure 4), they do not mention what happens to the Slc38a5-/- retina at P12 immediately after the hyperoxia phase. Is the vaso-obliteration altered in the Slc38a5-/- retina at that time compared to the WT?

7) What happens to the neurovascular unit (pericyte, astrocyte, Müller glia etc) in the Slc38a5-/- retina? How do they respond to altered angiogenesis?

8) Overall, the Discussion needs to emphasize the role of endothelial cell metabolisms in vascular development and maturation and how Slc38a5 may influence these processes.

*Reviewer #2 (Recommendations for the authors):*

1) The authors show western blots immunostained to detect SLC38A5 protein. While the Lrp5 blot image in Figure 1E is consistent with its graph in Figure 1F, the blot measuring SLC38A5 in Ndpy/+ and Ndpy/- samples are not convincing. If another blot was used to prepare the data in Figure 1F, then please include it here. Their finding that SLC38A5 has a much more dramatic drop in RNA levels than the corresponding protein levels is intriguing (Figure 1A, B versus Figure 1E, F). This same phenomenon is seen when transfecting human endothelial cells with the inhibitory siRNA (Figure 5A versus 5B). I would welcome the authors to include a brief explanation of this result in the Discussion section.

2) Evidence for a direct interaction between Wnt signaling and protein binding on DNA regulatory elements is missing. The authors identified three putative TCF binding regions upstream of the Slc38a5 gene and cloned these into luciferase expression vectors for use in standard reporter gene assays. In Figure 2D, they present data showing significant response of their luciferase containing constructs to activated β-catenin. As each region identified is over 500 bp, a concern is that the β-catenin is having an indirect effect; that is, regulating another gene(s) that then regulate Slc38a5. An additional experiment with a construct containing mutant Wnt binding site(s) would be evidence of direct binding. Also, please include the replicate number of transfections per experiment performed for the luciferase assay in the Methods or figure legend.

3) The authors performed very thoughtful and thorough experiments to determine that SLC38A5 transports glutamine, which is required for endothelial cell viability, migration and tubular formation (Figure 6). To block glutamine uptake by the endothelial cells, they used glutamine antagonist, DON. To measure cell viability, they treated cells at 0.1 mM DON, yet to measure migration, they used five times the concentration (0.5 mM) and for tubular formation, they used ten times that amount (1 mM). I am concerned that the cell viability was compromised at these higher concentrations of DON. Did they account for cell viability/death at these higher DON levels?

*Reviewer #3 (Recommendations for the authors):*

My detailed concerns are as follow.

1. Figure 1A, B: Data showed clear downregulation of SLC38A5 in both Lrp5-/- and Ndpy/- retinas, but the control groups seemed different. Please explain why the WT controls showed different expression patterns in Figure 1A and Figure 1B.

2. Figure 3 Does the Slc38a5 HET mice show any impaired retinal development?

3. Figure 4: The OIR model shows decreased levels of Slc38a5 at P8 and P12, which is the vaso-obliteration phase of OIR. Have the authors investigated the functions of Slc38a5 in this phase?

4. There are 2 LCM data, Figure 1C and Figure 4B. It seems in Figure 1C, the authors collected all three layers of vasculature but in Figure 4B, only collected the superficial layer. Please confirm and if it is so, have the authors checked if there is a difference regarding the expression levels of Slc38a5 in the superficial layer and all layers?

5. Figure 4: While the authors show that there is a decrease in pathological neovascularization in the Slc38a5-/- retina at P17 in the OIR model, they never mention what happens to the Slc38a5-/- retina at P12 immediately after the hyperoxia phase. Is the vaso-obliteration altered in the Slc38a5-/- retina at that time compared to the WT?

---

## [Author Response]

Please address all of the concerns of the reviewers. These were seen as Essential Revisions:1) The conclusion in Figure 1 is that Slc38a5 expression is downregulated in Lrp5 and Ndp knockouts. However, controls used in 1A and B are confusing. Please establish better temporal expression of the transporter in control (WT) retinas. Furthermore, While it's clear that Slc38a5 mRNA and protein expression is enriched in LCM-isolated retinal vessels, it's unclear whether that expression is exclusively in ECs or also in vessel associated mural cells (Figure 1C, Figure S1). Although Figure S1 shows the mining of mouse retinal scRNA-seq database to demonstrate exclusive Slc38a5 expression in ECs, please validate that in the tissue using either RNA in situ hybridization or IHC.

We thank the reviewer for the suggestion. For the expression of *Slc38a5*, the confusion and inconsistency were caused primarily by the large technical variation in WT retinas for LRP5 KO group. We have repeated time course expression for *Slc38a5* mRNA in Figure 1 and the updated figure is now included in Figure 1A . Please note that while both Lrp5 KO and Ndp KO mice are of C57BL/6J background, their controls are from different breeding colonies and hence some differences in their gene expression levels are expected between colonies.

For Slc38a5 expression in ECs, we have performed IHC in WT retinal cross sections that confirmed the colocalization of SLC38A5 antibody (green) with isolectin (red), which stains blood vessel endothelial cells. These results are now incorporated as the new Figure 1D and described in the Results section.

2) The conclusion in Figure 2 that Slc38a5 is a direct target of Wnt signaling is based on correlative data, yet a direct interaction (between β-catenin protein and the DNA driving the regulation of Slc38a5 to the retina) is not shown. They are assuming that the 5' flanking regions upstream of Slc38a5 drive retina-specific expression. Another assumption is that the addition of β-catenin is directly binding to these regions. An additional experiment with a construct containing mutant Wnt binding site(s) would be evidence of direct binding.

We thank the reviewer for this suggestion and have tried to perform the suggested experiment containing mutated β-catenin binding sites to determine whether it is direct binding. While we consistently observed significantly increased β-catenin-driven luciferase activity by DNA regions upstream of *Slc38a5* gene, however, with mutated DNA regions, we are not able to detect consistent abolishment of reporter activity after multiple repeated experiments. Overall the mutated constructs produced varying results unable to support a definitive conclusion. In response to reviewer #2’s concern and suggestions on indirect binding, we have therefore toned down the conclusion from Figure 2, and now state that *Slc38a5* is a downstream target regulated by Wnt signaling, yet whether this downstream regulation is a direct transcriptional target is unclear. Regulation by an indirect, secondary mechanism is also possible, such as hypoxia (see results and discussion). Constructs of shorter DNA regions containing binding sites will be considered in future studies.

3) Figure 3 has low quality images with data not matching the images, please address.

The low image quality of Figure 3 was caused by the Word to PDF conversion process. High resolution figures in PDF are now uploaded separately for the revision. Figure 3D was also updated with more representative images matching the quantified results.

4) The conclusion that the blood vessels in the adult are normal (Figure S2) needs to be substantiated with data.

We have now quantified the blood vessel density in the adult retinas, which shows comparable levels of vascular coverage, the numbers of junctions and branches in WT vs. Slc38a5 KO retinas. This data is now included as a new supplemental figure (Figure 3—figure supplement 2) and in the results.

5) A lack of data at P12 in the OIR model is an issue.

We have evaluated WT vs. Slc38a5 KO retinas at P12 in OIR and quantified the levels of vaso-obliteration (VO), which shows comparable VO levels in WT vs. Slc38a5 KO retinas at P12. This data is now included as a new supplemental figure (Figure 4—figure supplement 1).

6) Cell viability and use of DON with varying concentrations, please address.

Varying concentrations of DON was used to suit each experiment’s goal and purpose. We did perform dose-dependent assay for cell viability (MTT) with varying dose of DON (0-1mM) and Gln (2mM), and it didn’t show significant dose-dependency on DON concentration. Nevertheless DON also didn’t show significantly increased toxicity at higher dose (0.5-1mM) compared with lower doses. Which is why it was used at higher concentration for other assays (also see point #7).

7) The authors conclusion that glutamine uptake is essential for cell viability, migration and tubular formation is based on data shown in Figure 6. There is a concern that cell viability is affecting cell migration and tubular formation, as the glutamine inhibitor is used at a much higher concentration for the latter two assays in this figure.

As discussed in response above, cell viability was not significantly affected at higher concentration compared with lower concentration of DON, as up to 1mM of DON didn’t show severe toxicity. In addition, for migration and tubular formation, mitomycin was added to inhibit cell proliferation, therefore the difference we observed in migration and tubular formation was unlikely due to differences in cell viability/proliferation caused by DON.

8) Overall, the Discussion needs to emphasize the role of endothelial cell metabolism in vascular development and maturation and how Slc38a5 may influence these processes.

Thank you for the suggestions. The discussion was revised accordingly to emphasize EC metabolism in angiogenesis and the role of Slc38a5 in influencing glutamine (and other AA) availability in this process.

Reviewer #1 (Recommendations for the authors):The study focuses on the role of SLC38A5, a neutral amino acid transporter, in retinal angiogenesis. The authors show that Slc38a5 transporter is highly enriched in normal retinal vascular ECs, and upregulated in the ECs in pathogenic neoangiogenesis (the OIR model). Additionally, the authors show that Slc38a5 transcription is regulated by Wnt/β-catenin signaling and deletion of Slc38a5 in mice substantially delays retinal vascular development and suppresses pathological neovascularization in the OIR model by suppressing glutamine uptake and reducing VEGFR2 expression. The authors claim that SLC38A5 is a new metabolic regulator of retinal angiogenesis.The study is performed carefully and demonstrates clearly an important role for the transporter in retina angiogenesis. However, there are some concerns that need to be addressed as follows:1) The authors show that Slc38a5 is downregulated in the Lrp5-/- and Ndpy/- retinas (Figure 1A, B); however, there is a discrepancy in Slc38a5 expression levels in the control retinas. The expression of Slc38a5 in the WT retina goes down from P8-P12 and then plateaus through P17 (Figure 1A). In contrast, in Figure 1B, the expression of Slc38a5 in the Ndpy/+ retina plateaus from P8-P12 and then goes up through P17. The authors need to establish better the temporal expression of the transporter in control (WT) retinas.

We thank the reviewer for the suggestion. Please refer to Essential Revisions #1 for response.

2) While it's clear that Slc38a5 mRNA and protein expression is enriched in LCM-isolated retinal vessels, it's unclear whether that expression is exclusively in ECs or also in vessel associated mural cells (Figure 1C, Figure S1). Although Figure S1 shows the mining of mouse retinal scRNA-seq database to demonstrate exclusive Slc38a5 expression in ECs, it's necessary to validate that in the tissue using either RNA in situ hybridization or IHC for in combination with an endothelial cell or mural cell marker.

We thank the reviewer for the suggestion. Please refer to Essential Revisions #1 for response.

3) Figure 3: The image qualities are poor. The authors need to enhance image qualities to show the vessels clearly in such low magnification.

We thank the reviewer for the suggestion. Please refer to Essential Revisions #3 for response.

4) Figure 3F: The images in this panel show more than 50% decrease in the vascular area in the deep plexus between WT and Slc38a5-/- retinas. However, the graph shows a far lower (10-15% at best) decrease in the vascular coverage. The authors need to select representative images to match the graph.

Figure 3F does not show more than 50% decrease in vascular area. We assume the reviewer is referring to Figure 3D, which is now updated with more representative images. Please also refer to Essential Revisions #3 for response.

5) The authors show the presence of vessels in the adult Slc38a5-/- retina to claim that vascular abnormalities seen in early development are gone in the adult (Figure S2). However, the presence of vessels does not mean that there are no vascular abnormalities. The authors should compare established vascular parameters such as branching-density, vascular pruning between adult WT and Slc38a5-/- retinas to justify the claim.

Please refer to Essential Revisions #4 for response.

6) While the authors show that there is a decrease in pathological neovascularization in the Slc38a5-/- retina at P17 in the OIR model (Figure 4), they do not mention what happens to the Slc38a5-/- retina at P12 immediately after the hyperoxia phase. Is the vaso-obliteration altered in the Slc38a5-/- retina at that time compared to the WT?

Please refer to Essential Revisions #5 for response.

7) What happens to the neurovascular unit (pericyte, astrocyte, Müller glia etc) in the Slc38a5-/- retina? How do they respond to altered angiogenesis?

In adult WT and *Slc38a5^-/-^* retinas, GFAP staining showed comparable levels of staining pattern around ganglion cell layer (RGC), potentially reflecting astrocyte staining. There is no detectable sign of Müller glia activation, which would have been reflected by GFAP positive Müller glial processes. A more definitive conclusion regarding the neurovascular unit can be drawn by specific staining and thorough quantification with other cell specific markers (eg. NG2 for pericyte and glutamine synthetase for Müller glia), as well as GFAP in retinal flat mounts. These experiments are beyong the scope of our current study and will further evaluate in future studies when feasible.

8) Overall, the Discussion needs to emphasize the role of endothelial cell metabolisms in vascular development and maturation and how Slc38a5 may influence these processes.

The discussion is now revised to reflect this suggestion, also see response to Essential Revisions #8.

Reviewer #2 (Recommendations for the authors):1) The authors show western blots immunostained to detect SLC38A5 protein. While the Lrp5 blot image in Figure 1E is consistent with its graph in Figure 1F, the blot measuring SLC38A5 in Ndpy/+ and Ndpy/- samples are not convincing. If another blot was used to prepare the data in Figure 1F, then please include it here. Their finding that SLC38A5 has a much more dramatic drop in RNA levels than the corresponding protein levels is intriguing (Figure 1A, B versus Figure 1E, F). This same phenomenon is seen when transfecting human endothelial cells with the inhibitory siRNA (Figure 5A versus 5B). I would welcome the authors to include a brief explanation of this result in the Discussion section.

We thank the reviewer for the opportunity to clarify this point. The reason that SLC38A5 protein levels in *Ndp^y/+^* and *Ndp^y/-^* samples in Western appear less dramatic as the LRP5 blot is due to the loading control levels, where the sample loading (as seen for GAPDH) is less in *Ndp^y/+^* than *Ndp^y/-^* group, resulting the visual perception of less drastic and underwhelming difference between groups. Yet after normalization of densitometry that accounted for the difference in loading controls, the actual difference is reflected in the quantified bar graphs (Figure 1F), which was prepared from the images shown in Figure 1E.

Regarding why the RNA levels are more drastic than the protein levels, it has been reported in many previous studies that correlation between mRNA and protein levels are quite poor, with only ~40% explanatory power (de Sousa Abreu, R., et al., Global signatures of protein and mRNA expression levels. Mol. Biosyst. 2009; Vogel, C. et al., Insights into the regulation of protein abundance from proteomic and transcriptomic analyses. Nat. Rev. Gen., 2012; Antonis Koussounadis, et al., Relationship between differentially expressed mRNA and mRNA-protein correlations in a xenograft model system. Scientific Reports, 2015). This discrepancy has often been attributed to other regulatory mechanisms which differ between mRNA transcripts and proteins, such as post-translational modification/regulation and degradation (Maier, T. et al., Correlation of mRNA and protein in complex biological samples. FEBS Lett., 2009). A brief explanation on this point is included in the discussion.

2) Evidence for a direct interaction between Wnt signaling and protein binding on DNA regulatory elements is missing. The authors identified three putative TCF binding regions upstream of the Slc38a5 gene and cloned these into luciferase expression vectors for use in standard reporter gene assays. In Figure 2D, they present data showing significant response of their luciferase containing constructs to activated β-catenin. As each region identified is over 500 bp, a concern is that the β-catenin is having an indirect effect; that is, regulating another gene(s) that then regulate Slc38a5. An additional experiment with a construct containing mutant Wnt binding site(s) would be evidence of direct binding. Also, please include the replicate number of transfections per experiment performed for the luciferase assay in the Methods or figure legend.

Please see response to Essential Revision #3 for detail. We have now revised and toned down the conclusion to include the possibility of an indirect effect by a second regulatory mechanism (see results and discussion). The number of reporter experiments repeated was now included in the methods and figure legends, where at least three rounds of experiments were performed for the luciferase assays, and for each experiment each transfection group has at least 5 replicates.

3) The authors performed very thoughtful and thorough experiments to determine that SLC38A5 transports glutamine, which is required for endothelial cell viability, migration and tubular formation (Figure 6). To block glutamine uptake by the endothelial cells, they used glutamine antagonist, DON. To measure cell viability, they treated cells at 0.1 mM DON, yet to measure migration, they used five times the concentration (0.5 mM) and for tubular formation, they used ten times that amount (1 mM). I am concerned that the cell viability was compromised at these higher concentrations of DON. Did they account for cell viability/death at these higher DON levels?

Please see response to Essential Revision #6 for detail. Cell viability did not show clear dose-dependence between 0.1-1mM Don. The doses were performed at different time and were chosen to suit each experiment’s needs. Although cell viability showed slight decrease from 0.1mM to 0.5 and 1mM, we do not expect the modest difference is attributable to the observed differences of cellular migration and tubular formation assays, since the assays were also performed with mitomycin to inhibit cell proliferation.

Reviewer #3 (Recommendations for the authors):My detailed concerns are as follow.1. Figure 1A, B: Data showed clear downregulation of SLC38A5 in both Lrp5-/- and Ndpy/- retinas, but the control groups seemed different. Please explain why the WT controls showed different expression patterns in Figure 1A and Figure 1B.

Please see response to Essential Revision #1 for detail.

2. Figure 3 Does the Slc38a5 HET mice show any impaired retinal development?

The Slc38a5 het mice do not show significantly impaired retinal development compared with WT.

3. Figure 4: The OIR model shows decreased levels of Slc38a5 at P8 and P12, which is the vaso-obliteration phase of OIR. Have the authors investigated the functions of Slc38a5 in this phase?

Please see response to Essential Revision #5 for details on P12 OIR data.

4. There are 2 LCM data, Figure 1C and Figure 4B. It seems in Figure 1C, the authors collected all three layers of vasculature but in Figure 4B, only collected the superficial layer. Please confirm and if it is so, have the authors checked if there is a difference regarding the expression levels of Slc38a5 in the superficial layer and all layers?

Figure 1C showed normal retinal vasculature with three layers of blood vessels. In Figure 4B in OIR, however, only superficial layer of vessels are present and the deep vascular layer development is hindered and absent in OIR retina. Therefore only superficial layer of vessels were collected and compared in Figure 4B.

Currently it is unclear whether superficial vs. deep layer of vessels show differential expression levels of Slc38a5, although our immunohistochemistry data (new Figure 1D) demonstrated positive staining in all three vascular layers, suggesting at least it is present in all vessel layers.

5. Figure 4: While the authors show that there is a decrease in pathological neovascularization in the Slc38a5-/- retina at P17 in the OIR model, they never mention what happens to the Slc38a5-/- retina at P12 immediately after the hyperoxia phase. Is the vaso-obliteration altered in the Slc38a5-/- retina at that time compared to the WT?

Please see response to Essential Revision #5 for detail. P12 OIR analysis showed comparable VO levels between WT and KO.